

# Sub-seasonal variability of supraglacial ice cliff melt rates and associated processes from time-lapse photogrammetry

Marin Kneib[1,2], Evan S. Miles[1], Pascal Buri[1], Stefan Fugger[1,2], Michael McCarthy[1], Thomas E. Shaw[1], Zhao Chuanxi[3], Martin Truffer[4], Matthew J. Westoby[5], Wei Yang[3], Francesca Pellicciotti[1,5]

[1]High Mountain Glaciers and Hydrology Group, Swiss Federal Institute, WSL, Birmensdorf, 8903, Switzerland.
[2]Institute of Environmental Engineering, ETH Zürich, Zürich, 8092, Switzerland
[3]Key Laboratory of Tibetan Environment Changes and Land Surface Processes, Institute of Tibetan Plateau Research, Chinese Academy of Sciences, Beijing, 100045, China
[4]Geophysical Institute and Department of Physics, University of Alaska Fairbanks, Fairbanks, AK 99775, USA
[5]Department of Geography and Environmental Sciences, Northumbria University, Newcastle upon Tyne, NE1 8ST, UK

*Correspondence to*: Marin Kneib (marin.kneib@wsl.ch)

**Abstract.**

Melt from supraglacial ice cliffs is an important contributor to the mass loss of debris-covered glaciers. However, ice cliff contribution is difficult to quantify as they are highly dynamic features, and the paucity of observations of melt rates and their variability leads to large modeling uncertainties. We quantify monsoon season melt and 3D evolution of four ice cliffs over two debris-covered glaciers in High Mountain Asia (Langtang Glacier, Nepal, and 24K Glacier, Tibet) at very high resolution using terrestrial photogrammetry applied to imagery captured from time-lapse cameras installed on lateral moraines. We derive

weekly flow-corrected DEMs of the glacier surface with an estimated uncertainty of +/- 0.2 m for Langtang Glacier and +/- 0.06 m for 24K Glacier and use change detection to determine distributed melt rates at the surfaces of the ice cliffs throughout the study period. We compare the measured melt patterns with those derived from a 3D energy balance model to derive the contribution of the main energy fluxes. We find that ice cliff melt varies considerably throughout the melt season, with maximum melt rates of 5 to 8 cm.day⁻¹, which is 3 to 38 times higher than the melt rates of the surrounding debris-covered

ice. Our results highlight the influence of redistributed supraglacial debris on cliff melt. At both sites, ice cliff albedo is influenced by the presence of thin debris at the ice cliff surface, which is largely controlled on 24K Glacier by liquid precipitation events that wash away this debris. Slightly thicker or patchy debris reduces melt by 1-2 cm.day⁻¹ at all sites. Ultimately, our observations show a strong variability in cliff area, which is controlled by supraglacial streams and ponds and englacial cavities that promote debris slope destabilization and the lateral expansion of the cliffs. These findings highlight the

need to better represent processes of debris redistribution in ice cliff models, to in turn improve estimates of ice cliff contribution to glacier melt and the long-term geomorphological evolution of debris-covered glacier surfaces.



# 1 Introduction

Ice cliffs are one of the main contributors to the mass loss of debris-covered glaciers and are likely to contribute to the 'debris-cover anomaly', which describes the tendency of debris-covered glaciers to display similar ablation rates to clean ice glaciers

at the same elevation despite the insulating effect of debris (Gardelle et al., 2013; Pellicciotti et al., 2015; Buri et al., 2021). Along with supraglacial ponds, ice cliffs are directly exposed to energy fluxes from the atmosphere and therefore act as 'melt hotspots' relative to the surrounding debris-covered ice (Steiner et al., 2015; Buri et al., 2016a; Miles et al., 2016). Indeed, beyond a few centimeters of debris, melt rates reduce exponentially with increasing debris cover thickness (Ostrem, 1959; Nicholson and Benn, 2006; Reid and Brock, 2010). A series of studies based on high-resolution remote sensing data acquired

from unoccupied aerial vehicles (UAVs) and satellite sensors have shown that ice cliffs enhance melt by a factor of 1.2 to 14 (Immerzeel et al., 2014; Juen et al., 2014; Mölg et al., 2019; Thompson et al., 2016; Brun et al., 2018; Mishra et al., 2021; Reid and Brock, 2014) and ponds by a factor of 4 to 8 (Stefaniak et al., 2021; Salerno et al., 2017) relative to their surrounding debris-covered area. Similarly, modeling studies using advanced energy balance models at the scale of an entire glacier or catchment have estimated the melt enhancement factors to be between 6 and 13 for ice cliffs (Buri et al., 2021) and between 9

and 17 for ponds (Miles et al., 2018).

Both remote sensing and modeling based approaches to quantify ice cliff melt have limitations. Remote sensing approaches typically focus on deriving melt estimates from 'hot spots' of high thinning identified in maps of elevation change, and which need to be corrected to account for glacier flow (Vincent et al., 2016; Brun et al., 2018; Miles et al., 2018, 2021; Mishra et al., 2021). However, attributing the melt to the cliffs is non-trivial as they are particularly difficult to map from remote sensing

data, either manually, or using automated methods (Herreid and Pellicciotti, 2018; Kneib et al., 2020). Once the cliff outlines at the start and the end of a focus period are known, there are various ways of extrapolating the melt between the two digital elevation models (DEMs) that may lead to varying results (Brun et al., 2016; Mishra et al., 2021), while the cliff outlines may have varied considerably within a few months (Watson et al., 2017b). Wet and cloudy conditions during the monsoon season, when ice cliffs are the most active, present additional challenges for acquiring time series observations of Himalayan debris-

covered glaciers using satellite sensors.

The modeling of the cliff energy balance is another way to tackle the problem of the cliff contribution to glacier melt. It has evolved in the past two decades from the point scale (Sakai et al., 2002; Han et al., 2010; Reid and Brock, 2014; Steiner et al., 2015) to a distributed representation of the energy balance at the cliff surface (Buri et al., 2016a). Accounting for the cliff energy balance to dynamically update the cliff geometry (Buri et al., 2016b) has led to a better understanding of the controls

of ice cliff evolution, including aspect (Buri and Pellicciotti, 2018), and to the estimation of ice cliff melt contribution at the catchment scale (Buri et al., 2021). This complex modeling framework is, however, still limited in the representation of the interaction of ice cliffs with their surroundings. For example, the model presented by Buri et al. (2016b) accounts for debris redistribution by removing debris on slopes solely based on a fixed slope threshold, and, for ice cliffs which are attached to a



pond, uses a fixed value of pond melt at the cliff base. Moreover, the model parameters have only been evaluated using a small
sample of cliffs, where data has been collected over short time-scales using ablation stakes, by measuring the backwasting rate
of the cliff edge (Sakai et al., 1998, 2002; Han et al., 2010; Reid and Brock, 2014; Steiner et al., 2015; Buri et al., 2016a), and,
more recently, using measured volume changes (Buri et al., 2016b). Ultimately, fully distributed energy balance models require
knowledge of meteorological and surface variables over the cliffs surface and the surrounding debris slopes, such as albedo,
which are difficult to determine, and which vary much in time and space.

These limitations highlight the need for detailed and quantitative observations of cliff melt and evolution during the melt
season, including observations of interactions with debris cover and supraglacial streams and ponds. This is particularly
challenging as with their high backwasting rates of several centimeters per day (Juen et al., 2014; Steiner et al., 2015), ice
cliffs are particularly dynamic features which can grow, shrink, appear or disappear within the course of a single season (Sato
et al., 2021; Kneib et al., 2021). Ice cliff area relative to the debris-covered area varies between 1 and 15% (Mölg et al., 2019;
Watson et al., 2017a; Steiner et al., 2019; Falaschi et al., 2021; Sato et al., 2021; Anderson et al., 2021), but this value regularly
changes by up to 20% from year to year (Kneib et al., 2021). This high variability can be explained by the strong influence of
local processes such as ponds (Kraaijenbrink et al., 2016; Watson et al., 2017b), streams (Mölg et al., 2020) and debris
redistribution (Moore, 2018; Westoby et al., 2020) leading to rapid ice cliff formation, splitting, merging, expansion and decay
(Kneib et al., 2021). To improve process understanding and, in turn, inform the refinement of numerical models, observations
of ice cliff evolution therefore need to: 1) be captured during the melt season, when ice cliff activity is at its highest; 2) be of
high spatiotemporal resolution and, in turn, 3) be suitable for quantifying surface changes both across ice cliffs, and on adjacent
local topography and features, including debris-covered ice, and supraglacial streams and ponds.

Time-lapse cameras are regularly used to study dynamic changes to the cryosphere: they have been set up at glacier calving
fronts (Minowa et al., 2018; How et al., 2019; Kneib-Walter et al., 2021), have been used to quantify fast glacier flow (Messerli
and Grinsted, 2015) and have been combined with numerical modeling to constrain snow melt and accumulation patterns
(Farinotti et al., 2010). In parallel, in the past decade, advances in modern 'structure-from-motion' (SfM) photogrammetry
have enabled the reconstruction of 3D topography from images acquired from multiple, converging viewing angles (Westoby
et al., 2012). A primary use of SfM-based approaches has been to map glacier surfaces from UAV, enabling the detailed study
of debris-covered glaciers and their supraglacial features (Immerzeel et al., 2014; Kraaijenbrink et al., 2016; Brun et al., 2018;
Westoby et al., 2020; Mishra et al., 2021). Topographic reconstruction can also be achieved through terrestrial
photogrammetric survey, which can enable the accurate mapping of steep and overhanging features which are common at ice
cliff locations (Brun et al., 2016; Watson et al., 2017b; King et al., 2020), and can be occluded in imagery acquired from nadir-
oriented aerial surveys. Terrestrial photogrammetric surveys have produced comparable results to more expensive and less
portable setups such as terrestrial laser scanners (TLS; Piermattei et al., 2015). The combination of high-frequency time-lapse
image capture and photogrammetric processing is therefore highly promising for generating quantitative observations of the



dynamics of fast-changing cryospheric landscapes. While still limited by the amount of processing required and the logistical aspects of deploying arrays of time-lapse cameras, time-lapse photogrammetry has been used successfully to precisely monitor thaw slump activity (Armstrong et al., 2018), lava flows (James and Robson, 2014b), snow melt (Filhol et al., 2019) and calving dynamics (Mallalieu et al., 2017).

Here we apply time-lapse SfM photogrammetry to study the subseasonal melt of four ice cliffs on two different glaciers of the Himalayan range, at weekly intervals during a full melt season. **We aim to quantify ice cliff subseasonal melt and identify the local processes controlling its variability.** To this end, we derive weekly flow-corrected DEMs of the ice cliffs and calculate spatially distributed melt over the study period. We compare these results with estimates of melt generated by a 3D energy balance model to isolate the main energy fluxes and identify the local processes that cause modeled melt to deviate
from our measurements.

## 2 Data

### 2.1 Study sites

We installed time-lapse camera arrays on two Himalayan debris-covered glaciers with distinct glaciological and climatic characteristics (Fig. 1). Langtang Glacier is located in Central Nepal (85.72°E, 28.27°N) and has a 15 km-long debris-covered
tongue, with an estimated density of supraglacial ice cliffs (ponds) ranging between 2.1% and 4.7% (0.9% and 2.5%) of the debris-covered area (Kneib et al., 2020, 2021; Steiner et al., 2019; Miles et al., 2017b). The debris thickness increases down-glacier and exceeds 2 m in the lower portion of the glacier (McCarthy et al., 2021), where our survey domain was located (Fig. 1c). 24K Glacier (hereafter '24K') is located in Eastern Tibet (95.72°E, 29.76°N), is also extensively debris-covered but is much smaller than Langtang Glacier; it has a 2 km-long debris-covered tongue, and debris cover is thinner (at most 0.5 m in
the lower portion of the glacier) (Fig. 1d). 24K is much steeper (31.5° slope) than Langtang (15.6°), which may partly explain the scarcity of ponds at its surface, and the presence of a number of supraglacial streams in its central area which have led to the development of so-called 'cryo-valleys' bounded by ice cliffs, and similar to those described on Zmutt Glacier in Switzerland (Mölg et al., 2020).

We installed an array of eight time-lapse cameras on the lateral moraine of Langtang Glacier, overlooking a small domain of the lower portion of the debris-covered tongue and comprising a number of North-East to North-West facing ice cliffs, three of which are connected to a pond (Fig. 1a). An additional array of four time-lapse cameras was installed on 24K, overlooking a large stream-influenced North-facing cliff (Fig. 1b).





## 2.2 Time-lapse camera arrays

The time-lapse cameras were mounted directly on stable boulders along lateral moraine crests of the two glaciers (Fig. 1a, b). The custom time-lapse rigs consisted of a Canon EOS 2000D camera (24.1 MP) with an 18-25 mm lens. The cameras took photographs at a consistent 2-hour interval for the whole duration of the melt season, triggered by an intervalometer. The cameras were powered by a 5W solar panel, a 12V 7Ah lead-acid battery and an ECO-N-T solar charge controller (Fig. 2a). These elements were assembled in a weather-proof box (Bixibox) which was mounted on a 1.5 m-long aluminum mast bolted

vertically to the rock (Fig. 2b, c). The total cost of one setup was ~1900 €. All camera stations ran without data gaps from 12/05/2019 to 01/11/2019 (6 months) for Langtang, and from 08/06/2019 to 12/10/2019 (4.1 months) for 24K.The focal length of all cameras was manually set to 18 mm to ensure the widest viewing angle. The xyz position of each time-lapse camera was measured using a differential GPS (dGPS), and the three viewing angles were measured at the beginning and at the end of the time-lapse period.

## 2.3 UAV flights and remote sensing imagery

We carried out a UAV survey of the study domain at the start and the end of the monitoring period on 24K. On Langtang, a flight was only possible at the end of the study period (Table 1). The initial and final conditions for Langtang were instead constrained with two Pléiades stereo images taken within a month of the start and the end of the time-lapse recording period (Table 1).


The UAV images were taken nadir-oriented at a fixed elevation of 70 to 120 m above the glacier surface with a lateral overlap of 70% and a forward overlap of ≥80%. Additional oblique images of the survey domain, which have been proven to mitigate against the introduction of systematic model deformation (James and Robson, 2014a), were taken manually, depending upon UAV battery limits.


Between 15 and 18 ground control points (GCPs) were laid out across the survey domain around the main features of interest (ice cliffs, ponds and streams), with a good distribution between topographic lows and highs (Fig. 4), and consistent coverage at margins of the study area. GCPs were visible in photographs captured by the time-lapse cameras, and from the UAV. The xyz positions of the GCPs were measured with a single-band dGPS system (10 cm accuracy) within 48 hours of the UAV

flights.

## 2.4 GPR measurements

We conducted ice thickness measurements using a Kentech ground penetrating radar (GPR) monopulse generator with 20 m dipole antennas (~2.5 MHz) along four transects on 24K Glacier in October 2019 (Fig. 1b). For Langtang, we used the measurements from Pritchard et al. (2020) conducted in the vicinity of the survey domain (Fig. 1a). These measured ice

thicknesses were used to bias-correct the consensus ice thicknesses from Farinotti et al. (2019) using a linear regression of the ice thickness for Langtang and of the bed altitude for 24K to obtain a distributed estimate of ice thickness for each survey area. These corrections led to the reduction of the mean bias from 15.8 to 0.1 m and from 94.1 to 1.7 m for Langtang and 24K, respectively.

**2.5 Field observations of supraglacial ponds**

Two of the cliffs in the Langtang survey domain had a pond at their base in May 2019, at the start of the recording period. We monitored the two pond water level changes using HOBO pressure transducers and recorded the water surface temperature using a HOBO thermistor attached to a float. The pond at the base of one cliff drained almost entirely during the study period and it was not possible to retrieve its pressure transducer, which got buried by a thick layer of debris. The thermistor was, however, still accessible and its temperature record combined with the observations from the time-lapse cameras clearly shows

the timing of the drainage.

**2.6 Meteorological observations**

Each glacier was equipped with an on-glacier automatic weather station (AWS), which was installed in the vicinity of the survey domain (<100 m of elevation difference) and recorded, among other variables, air temperature, relative humidity, incoming and outgoing longwave and shortwave radiation and wind speed at 5 minute intervals over the study period (Fugger

et al., 2022). Precipitation measurements were acquired using a HOBO tipping bucket at the AWS site for 24K and on the lateral moraine, ~500 m away from the AWS, for Langtang (Steiner et al., 2021). The air temperature measured at the AWS location was lapsed considering the mean above-debris lapse rates (-0.0088°C m$^{-1}$) following Shaw et al. (2016). All other variables were left unadjusted for input to the energy balance model (Section 5.7).

**3 Methods**

**3.1 Processing of UAV and Pléiades images**

The Langtang Pléiades satellite images were stereo-processed to generate 2 m-resolution DEMs and 0.5 m-resolution orthoimages from the panchromatic band using Rational Polynomial Coefficients (RPCs) within the NASA AMES Stereo Pipeline (Kneib et al., 2020; Beyer et al., 2018; Shean et al., 2016).

The Langtang and 24K UAV images were imported to Agisoft Metashape Professional (v1.7.2). Initial bundle adjustment was performed using only the UAV GPS geotags. We then incorporated the xyz positions of the GCPs to refine this adjustment and improve camera location and pose estimation, and the location of image tie points (Westoby et al., 2020). We then generated dense point clouds, which were used to produce DEMs and orthoimages (0.2 m resolution for the Langtang survey, 0.12 m for the 24K survey).




Co-registration of the Langtang Pléiades DEMs was performed over off-glacier stable terrain with slopes between 10 and 45°, following the approach detailed in Nuth and Kääb, (2011). For the 24K UAV flights, we used the fixed position of the time-lapse cameras, which we measured during each dGPS survey, to correct for vertical and horizontal shifts in the position of the on-glacier GCPs. After the initial co-registration of the UAV DEMs, there remained some non-linear distortions (tilts) that

were removed using additional natural off-glacier control points (boulders) on both sides of the glacier from the June flight to rerun the bundle adjustment of the October flight, which improved the co-registration.

For both sites we estimated a global vertical uncertainty by calculating the DEM difference over off-glacier stable terrain (Mishra et al., 2021). For the Langtang Pléiades DEMs, the mean bias (+/- standard deviation) was 0.10 (+/- 0.53 m). For the

24K UAV DEMs, the mean bias was -0.38 (+/- 0.52 m).

### 3.2 DEM processing with time-lapse photogrammetry

The overall workflow for generating DEMs from the time-lapse images broadly follows that described by Mallalieu et al. (2017) (Fig. 3). The time-lapse lasted from 12/05/2019 to 01/11/2019 (173 days) for Langtang, and from 08/06/2019 to 12/10/2019 (126 days) for 24K, resulting in ~2100 images per camera for Langtang and ~1550 images per camera for 24K, at

24.1 MP resolution.

In the first step (step 1, Fig. 3) we manually removed all the images taken during night time, with water or snow in front of the lens, or with poor visibility due to clouds or precipitation. We then grouped the images from the different cameras taken at each site within 2.5-hour periods to account for offsets in the image acquisition time of the different cameras. If one or more

images were missing, we ignored all the images in that set. After this pre-selection, there remained 781 image sets for Langtang and 357 for 24K.

We used the image sets of 01/11/2019 14:00 for Langtang and 08/06/2019 14:00 for 24K as 'reference' sets, as they were taken within a few hours of the GCP surveys conducted in the survey domain (Fig. 4). In step 2, the reference image sets were

imported to Agisoft Metashape Professional (v1.7.2) and we used the dGPS-measured position and viewing angles of the cameras in the initial bundle adjustment and then used the GCP coordinates for subsequent optimization of the lens parameters prior to generation of the dense point cloud (Fig. 3). From these reference sets we exported all optimized camera information (xyz position, pose and lens parameters) as well as the camera-specific, pixel-based locations of a series of 'pseudo' GCPs (PGCPs) which took the form of boulders and other distinctive features located on stable background terrain (Fig. 4). We used

21 PGCPs for Langtang and 14 for 24K. A majority of these PGCPs were located on the inside of the opposite lateral moraine of each glacier, as this was the closest 'stable' terrain to the survey domains. We were careful to select boulders that showed limited movement by ensuring that the distance in the images between the boulder and crest did not change over the time-lapse



period. This therefore limited the possible influence of moraine collapse or slumping on the robustness of these features in the SfM workflow (Van Woerkom et al., 2019).


The accuracy of the PGCP position and camera parameters (location and pose) are important for the uncertainty of the final results. We optimized these accuracy values to minimize the bias in elevation over background stable terrain (Table S1, S2). As an initial guess, we used initial estimates provided by Agisoft Metashape (for PGCP position accuracy and lens parameters) and measurements of camera positions (with dGPS) and viewing angles conducted at the start and the end of the survey periods.

The position of the camera on the mast did not change by more than five centimeters, and the camera pose parameters by less than 5°, although this may have been temporarily exceeded during the observation period if wind speeds were very high.

In step 3 (Fig. 3), all the weekly image sets were processed semi-automatically in Agisoft Metashape, using the Agisoft Python API, following a 3-step workflow. In the first (fully automated) step the image sets were imported, along with the reference camera parameters, and underwent an initial bundle adjustment and camera lens parameter optimization without PGCPs. In a

second (manual) step, the PGCP positions and their associated accuracies were imported and the position of the PGCPs was manually adjusted in each image. In the third (fully automated) step the camera parameters were optimized after incorporating PGCP locations, and the final estimates of camera location and pose were used to build a final dense point cloud, which was then used to create a DEM and orthoimage.


We established the relative accuracy of the output DEMs by computing the mean and standard deviation values of elevation change relative to the reference DEM calculated over background stable terrain (Fig. 4, orange outline). The mean elevation change over this area of the background moraine was generally <0.2 m for Langtang and <0.05 m for 24K. Higher values were obtained when the illumination conditions in a given image set differed substantially from the reference image set, or when

some of the images were slightly blurred from rain or mist. In these instances, we used other, higher quality image sets taken within a few days from the target date.

Ultimately, we produced 25 time-lapse DEMs and orthoimages for Langtang (0.20 m resolution) and 19 DEMs and orthoimages for 24K (0.24 m resolution), covering the full study period at an approximately weekly interval (Table S3).


We used the orthoimages to manually delineate the ice cliff outlines at each weekly time-step, which we considered to be the exposed ice sections free of debris. This was sometimes difficult in the case of patchy debris, which was included in the cliff outlines when the underlying ice was still visible. Oblique viewing angles combined with a complex glacier surface led to gaps in the orthoimages and DEMs caused by topographic shadowing. Despite this limitation, we still resolved the larger portion of

three cliffs on Langtang and of the 24K cliff (Fig. 4a, b).



### 3.3 Glacier flow corrections

In a fourth step (step 4, Fig. 3), all the DEMs except a reference DEM for each glacier were corrected to account for glacier flow (horizontal surface velocity and emergence) following the approach described by Mishra et al. (2021), using estimates of distributed surface velocity and ice thickness to calculate the ice flux through a flux gate, and, in turn, an implied ice emergence
velocity. We calculated distributed surface velocity fields over the lower portion of the glaciers, including the survey domains, by applying a normalized cross-correlation approach to the Pléiades (for Langtang) and UAV (for 24K) DEM hillshades using ImGRAFT (Messerli and Grinsted, 2015). We filtered these velocity fields by removing values with a low signal-to-noise ratio (<2), low correlation score (<0.5) or unrealistically high values (>3 m for Langtang, >8 m for 24K over the study period) and interpolated the remaining results with a cubic spline interpolation (Mishra et al., 2021).

We used these velocity fields to correct the x-y displacements between the different DEMs (time-lapse, UAV, Pléiades) and the reference time-lapse DEMs, assuming a constant velocity over the study period. In this step we also accounted for vertical displacement due to the downslope advection of the surface using the slope from the AW3D 30 m-resolution DEM (Tadono et al., 2014) of each study area, smoothed using a 30-pixel Gaussian filter (Mishra et al., 2021; Miles et al., 2018; Brun et al.,
265   2018).

We calculated emergence velocity in the lower portion of the glaciers (including the survey domains) by estimating the flux through a flux gate located immediately upstream from the survey domains, taking into account the surface velocity and the adjusted ice thickness at this location (Mishra et al., 2021; Miles et al., 2018; Brun et al., 2018; Vincent et al., 2016) at a 8 m
resolution for Langtang, 4 m for 24K, and assuming that basal sliding accounts for 50% of the surface motion (but considered the full 0-100% range in the uncertainty calculation). We integrated the flux across the cross section with a simple-shear assumption to calculate the column-averaged velocity (Huss et al., 2007) and assumed that this flux is uniformly distributed as emergence downstream from the flux gate.

We estimated the surface velocity uncertainty as the normalized median absolute deviation of its x and y components over off-glacier terrain, equal to 0.84 m (0.6 cm.day$^{-1}$) for Langtang and 0.35 m (0.3 cm.day$^{-1}$) for 24K over the full study period. We obtained an emergence velocity of 0.39 +/- 0.16 m (0.3 +/- 0.1 cm.day$^{-1}$) for Langtang and 0.66 +/- 0.16 m (0.5 +/- 0.1 cm.day$^{-1}$) for 24K. As for the x-y displacements, we used these emergence values to correct the different DEMs (time-lapse, UAV, Pléiades) relative to the reference time-lapse DEMs assuming a constant emergence over time. Similarly to the DEMs, the cliff
outlines were flow-corrected for the surface displacements in the x and y directions.



### 3.4 Estimating melt from DEM differencing

In this study we were interested in calculating distributed melt patterns at the surface of the cliffs, which correspond to the normal displacement of the cliff surface (Buri et al., 2016b). A number of studies used the M3C2 algorithm (Lague et al., 2013; Watson et al., 2017b; Mishra et al., 2021) directly applied to the point clouds to calculate this normal displacement. However,

here we aimed to compare our results with a gridded ice cliff energy balance model (Buri et al., 2016a, b), which uses the cliff DEM for the distributed energy balance calculations. We therefore estimated the melt from two time-lapse DEMs (DEM1 and DEM2) by calculating for each pixel of DEM1 the local normal based on its eight neighboring pixels and finding the intersection of the normal with DEM2. The melt was then equal to the distance between the DEMs along this normal (Fig. S1). This approach is similar to the M3C2 algorithm but using DEMs and 3x3 neighborhoods.

### 3.5 Uncertainty estimation

In the case of two DEMs with the same slope parallel to one another, which we considered to be the most common short-term change due to ice melt, the elevation difference should be larger than the melt distances (e.g. Mishra et al., 2021), and the same should be true for their uncertainties. As the melt distance uncertainty was difficult to determine directly, we conservatively assumed the melt distance uncertainty to be equal to the elevation change uncertainty. In the case of our study areas with

complex geometries and viewing angles, we expected these uncertainties to vary with slope and aspect, as well as with the number of overlapping images, the distance from the time-lapse cameras, and the time difference with the reference DEMs (James and Robson, 2014b; Mallalieu et al., 2017; Armstrong et al., 2018; Filhol et al., 2019). As a minimum, we also expected the standard deviation of elevation change to increase with time from the reference image set and distance from the time-lapse cameras, except in the very near-field where less overlap of the images should lead to higher uncertainties (Mallalieu et al.,

300 2017).

We estimated the melt uncertainties in the cliff domain by analyzing the mean and standard deviation of elevation change over the moraine (Fig. 4). Indeed, the moraine was the closest feature to the survey domain that could be considered relatively 'stable', at least over a period of a few months. Furthermore, it had similar slopes and aspects to those of the cliffs in the survey

domain, but was located in the background of the survey area, making it a good but conservative proxy for the features analyzed (Fig. 4). We conducted two different tests to estimate the melt uncertainties in the cliff domain. The first test (1) was to look at the evolution of the mean and standard deviation of the elevation changes relative to the reference DEMs over the moraine with time (Fig, 5a, b). The second test (2) was to look at the evolution of the mean and standard deviation of the elevation changes with distance for time-lapse DEMs taken within a few days from each other (Fig. 5c, d). This enabled us to estimate

the $1\sigma$ melt uncertainties from the DEM differencing in the survey domain for individual pixels.



The mean value remained between +/-0.2 m for Langtang, where the moraine was ~800 m away from the cameras (Fig. 5a), and between +/-0.05 m for 24K, where the moraine was ~400 m away from the cameras (Fig. 5b). The standard deviation increased with time during the first two months of the time-series for Langtang, until it reached a value of ~1 m, while it

remained stable around 0.6 m for 24K during the whole period. For (2), we took the DEM the furthest away in time from the reference DEM and processed the image pairs taken within 48 hours of this new reference DEM, only keeping the resulting DEMs with a mean elevation change relative to the reference DEM lower than 0.2 m for Langtang (4 remaining DEMs) and 0.05 m for 24K (7 remaining DEMs) (Fig. 5a, b, dashed lines). The elevation change patterns of these near-contemporaneous DEMs highlighted a factor $f = 2$ increase in standard deviation with distance between the cliff domain and the moraine for

Langtang and $f = 1.7$ for 24K (Fig. 5c, d). As a result, we expressed the DEM uncertainty in the cliff survey domain as:

$$\sigma_{DEM} = \frac{\sigma_{Max}}{f},\tag{1}$$

where $\sigma_{Max}$ is the maximum standard deviation of the elevation change relative to the reference DEM over the moraine stable terrain. As a result, for Langtang $\sigma_{DEM} = 0.5\ m$ and for 24K, $\sigma_{DEM} = 0.4\ m$.

We also needed to account for the uncertainties related to the flow correction, which we assumed to be equal to the quadratic sum of the 1σ surface velocity uncertainty $\sigma_{xy}$, the 1σ emergence velocity uncertainty $\sigma_b$ estimated following the approach and assumptions described by Miles et al., 2018, and the uncertainty from the slope correction $\sigma_{Slope}$:

$$\sigma_{flow} = \sqrt{\sigma_{xy}^2 + \sigma_b^2 + \sigma_{Slope}^2}.\tag{2}$$

For the uncertainty on the slope correction, we assumed a 2° uncertainty in the slope angle. As a result, the 1σ uncertainty from flow correction was equal to 0.007 m.day$^{-1}$ for Langtang and 0.004 m.day$^{-1}$ for 24K.

The 1σ melt uncertainty for each pixel could be expressed as:

$$\sigma_{Melt} = \sqrt{\sigma_{DEM}^2 + \left(\sigma_{flow} \times dt\right)^2},\tag{3}$$

where $dt$ is the number of days over which the melt is calculated. When considering the melt of N pixels of the survey area and assuming a Gaussian error for independent measurements we obtained:

$$\sigma_N = \frac{\sigma_{Melt}}{\sqrt{N}},\tag{4}$$

Ultimately, we calculated melt on a tri-weekly basis for Langtang and a bi-weekly basis for 24K to reduce the uncertainties

relative to the measured melt-rates. This meant that the uncertainty from flow was an order of magnitude lower for these domains and led to an uncertainty in melt $\sigma_{DEM} = 0.5\ m$ for Langtang and $\sigma_{DEM} = 0.4\ m$ for 24K over their respective tri- and bi-weekly melt periods. Following the same approach with the maximum absolute mean values (0.2 m for Langtang and 0.05 m for 24K) and multiplying by a factor 2 to account for positive and negative biases, we constrained potential biases in



the vertical position of the time-lapse DEMs with (Eq. 1), to 0.2 m for Langtang and 0.06 m for 24K. According to Eq. 4, the
uncertainty from the standard deviation becomes negligible (<0.05 m) when the number of pixels N is greater than ~100 (300
for the 3σ uncertainty), however, this is not the case for a possible bias in the data that affects every pixel the same way.

### 3.6 Cliff brightness and snow events

For 24K we found the brightness of the cliffs to change substantially with time. We estimated this brightness for each set of
images (781 for Langtang, 357 for 24K), by taking the average value of the blue band in a 200 x 100 px domain at the center
of Cliff 1 (Langtang) and of the main 24K ice cliff (Fig. S2). We normalized this brightness value by the mean value of the
blue band in a domain of the same size over a debris-covered slope with similar slope and aspect characteristics, giving a basic
proxy for apparent changes in ice cliff albedo insensitive to illumination differences between scenes. We used the blue band
as when comparing the visible spectra of cliffs and debris of different brightnesses, this was the band that highlighted the
strongest differences. We took a single brightness value for all cliffs with different slopes and aspects in the domain, as the
brightness appeared to evolve in a similar way across all cliffs (Fig. S2).

We additionally looked at the daily influence of snow events on ice cliff melt at both sites. We considered that there was a
snow event when 1) the daytime shortwave albedo at the AWS location was higher than 0.3, or 2) when snowfall or fresh snow
cover on the glacier could be observed in at least one of the time-lapse images on a given day.

### 3.7 Energy balance model

We compared our tri- to bi-weekly melt patterns with the melt obtained over the same period using a static cliff energy balance
model (Buri et al., 2016a). The model calculates the energy inputs from shortwave, longwave and turbulent fluxes in a
distributed way across the cliff surface. It represents the state-of-the-art for the energy balance modeling of ice cliffs and has
been developed incrementally, successively including improved representation of the different fluxes in a more distributed
way (Sakai et al., 2002; Reid and Brock, 2014; Steiner et al., 2015; Buri et al., 2016a). Further developments included cliff
geometry changes and debris redistribution (Buri et al., 2016b), and this was used to demonstrate the persistence of north-
facing cliffs compared to south-facing ones (Buri and Pellicciotti, 2018) and ultimately estimate the total ice cliff melt
contribution to catchment-scale ice melt (Buri et al., 2021). The static version of the model that we used (Buri et al., 2016a)
has been described extensively in the past literature, which we invite the reader to refer to for further details (Steiner et al.,
2015; Buri et al., 2016a).

The model was run over the exact same periods over which we calculated melt from DEM differencing, without simulating
surface geometry changes. We used the static version of the model to focus on the contribution of the different energy-fluxes
only, thus removing the influence of the modeled geometry updates. We used the gap-filled time-lapse DEMs as the reference
surface over which to calculate the energy-fluxes and the debris-viewing angles and near-field horizon calculations. We filled



the gaps using the UAV DEMs corrected with the elevation change signal from the Pléiades (for Langtang) and UAV (for 24K) DEMs. To reduce computation time, the DEMs were resampled to 0.6m. We used the 30m AW3D DEMs (Tadono et al., 2014) of the area for the far-field horizon calculations and did not include debris redistribution or additional melt from the ponds.

380

We used data from the nearby on-glacier AWSs as meteorological forcing for the ice cliff model, and estimated debris surface temperature, which is an additional model input, from the outgoing longwave radiation at those AWSs. Importantly, we assumed a fixed albedo of 0.15 for debris and 0.2 for ice, which were the same values used in the original studies, calibrated on Lirung Glacier, which is located in the vicinity of Langtang Glacier (Steiner et al., 2015; Buri et al., 2016a).

## 4 Results

### 4.1 General measured and modeled melt patterns

#### 4.1.1 Site-scale melt patterns

Elevation change patterns between the pre- and post-monsoon period from the UAV and Pléiades DEMs alike showed enhanced surface lowering at the location of the ice cliffs (Fig. 6a, b). Elevation change patterns displayed some variability in the non-cliff area of the domain, and this was especially visible with the higher resolution data from 24K (Fig. 6b). Sub-debris melt on 24K, where the debris cover was thinner (Fig 1; McCarthy et al., 2021), also appeared to be higher than on Langtang (Fig. 6b). In both domains, cliff backwasting is evident (Fig. 6a, b), varying from cliff to cliff and site to site, between 0 and 5 cm.day$^{-1}$ on Langtang and 4 and 9 cm.day$^{-1}$ on 24K. There were also signs of cliff expansion (e.g. Cliff 3 on Langtang) and reburial (e.g. 24K main cliff). The mean sub-debris melt over the whole study period calculated from the flow-corrected Pléiades and UAV DEMs and for snow- and cliff-free (including a 5 m buffer around the initial and final cliff outlines) zones was -0.19 +/- 0.15 cm.day$^{-1}$ for Langtang and -1.2 +/- 0.5 cm.day$^{-1}$ for 24K (accounting for both uncertainties in the DEMs and melt variability), 6-38 (resp. 3-9) times less than the average melt at the cliff location measured with the time-lapse cameras over the same study period (Fig. 6).

The overall melt rate at the cliff location calculated from the Pléiades DEMs was 4% to 56% less than the total melt calculated from the time series of flow corrected time-lapse DEMs, while this difference was up to 27% between the UAV and time-lapse DEMs of 24K (Table S4).

To disentangle the different components of the ice cliffs' evolution, we focused the analysis of the sub-seasonal patterns on six transects of the three main Langtang cliffs and four transects of the 24K cliff, which kept similar aspect and slope during the whole study period (Fig. 6a, b), and were all west to north-east facing. The mean measured and modeled melt were





comparable for each of those transects, with ≤25% difference and no consistent bias, even though there was a higher variability in measured melt (Fig. 6c). The observed daily cliff melt was in general higher (3.9-5.1 cm.day$^{-1}$) for 24K than for Langtang (2.9-4.3 cm.day$^{-1}$). At the seasonal scale there did not appear to be a control of slope or aspect on melt (Fig. 6C, S3).

### 4.1.2 Melt patterns as a function of time

The time-lapse observations at both sites started a few days after the ice cliffs became snow-free, and ended after the first snowfalls. Overall, air temperatures were higher at 24K by 4-5 °C, but this difference was partly compensated by higher incoming shortwave radiation on Langtang (Fig. c, d). The incoming longwave radiations were of similar values, and plateaued during the whole monsoon season. Melt patterns at the two sites differed considerably. Melt was higher at the start of the study period (pre-monsoon; 3.4 +/-1.5 cm.day$^{-1}$ for Langtang, 6.7 +/- 2.1 cm.day$^{-1}$ for 24K) than at the end (post-monsoon; 0.7 +/- 1.1 cm.day$^{-1}$ for Langtang, 1.1 +/- 1.0 cm.day$^{-1}$ for 24K) (Fig. 7), and exhibited similar variability over the study period at both sites (coefficient of variation of 0.37 for Langtang Cliff 3, and 0.34 for the 24K cliff). The peak in melt was reached in the last week of June and first week of July on Langtang (6.4 +/- 1.9 cm.day$^{-1}$) and around mid-August on 24K (7.3 +/- 1.8 cm.day$^{-1}$). This peak in melt on Langtang corresponded with the timing of the peak in air temperature, but while air temperature stabilized between early July and early September, melt started to decrease from early July, coinciding with the decrease in incoming shortwave radiation and increase in longwave radiation at the start of the monsoon period (Fig. 7c). The peak in melt at 24K also corresponded to the maximum air temperature, but similar to Langtang, the incoming shortwave radiation had a direct influence on this melt pattern (Fig. 7d). The observed melt behaviors were well represented by the modeled melt but with slightly smaller amplitudes.

### 4.2 Processes occurring at each cliff

#### 4.2.1 Langtang Cliff 1

Langtang Cliff 1 was a relatively small (5 - 10 m tall, 30-40 m wide) north-facing cliff (Fig. 8, 9). In July it **expanded a few meters to the east but the new section got re-buried relatively quickly** in August: the small cavity at the base of the cliff (visible at the start of the study period, Fig. 8a) increased in size as the cliff backwasted, debris falling from the top of the cliff accumulated in the cavity slowly reburying this section of the cliff, which by the end of the period had become very shallow (Fig. 8f). Despite lateral expansion, the cliff lost 20% of its planimetric area during the study period (Fig. 9b), predominantly driven by an increase in average slope from 45° to 53° between mid-June and early August (Fig. 9e). Further, the mean aspect changed from 355°N to 325°N as the cliff expanded to west-facing slopes, but reverted to its original aspect with the reburial of this west-facing portion. The measured melt displayed a similar signal to that of Langtang Cliff 3 (Fig. 7a). It increased from 3.0 +/- 1.1 cm.day$^{-1}$ to 5.2 +/- 2.3 cm.day$^{-1}$ between mid-May and early July then displayed a slow decrease until the end of the study period when it reached the value of 0.0 +/- 0.7 cm.day$^{-1}$, similar to that of the surrounding debris-covered ice (Fig. 8f). The measured melt was relatively homogeneous across the cliff surface, except for higher values at the location of the cliff




expansion (Fig. 8h). The modeled fluxes showed a **strong contribution of shortwave radiation** to the cliff energy balance, with an **increase in the contribution of net turbulent fluxes from 9 to 36% between the end of May and mid-July when**
**the net shortwave decreased by 38% with the arrival of the monsoon** (Fig. 9d). The net longwave radiation contributed negatively to the cliff's energy balance.

### 4.2.2 Langtang Cliff 2

Langtang Cliff 2 was a medium size (10-20 m tall, 35-45 m wide) west-facing cliff that was attached to a pond at the start of the study period (Fig. 10, 11). The time-lapse images showed that the pond partly drained between 02/07 and 05/07. This was
confirmed by the pond surface temperature data (Fig. 11c), which showed much stronger temperature variations after the end of June, proof that the sensor had become grounded on the debris. The vertical step left by the pond at the cliff base after draining got progressively reburied and had disappeared by mid-August (Fig. 10c, d). The cliff had a concave shape at the start of the study period, being steeper at the top than at the bottom. End of June, **triggered by the cliff backwasting, part of the debris-covered slope above the cliff slumped, thus expanding the cliff upwards, at a lower angle** (Fig. 10, S6). This upper
debris-free area expanded laterally in July, enhancing the **sharp transition between the lower steeper portion that was progressively reburied after the drainage of the pond, and the upper shallower portion of the cliff that became predominant with time**. This reburial of the lower steeper section and expansion of the upper shallower section led to the cliff doubling in size in July and then returning to its initial size by early September.

For the analysis of the melt patterns, we focused on two north-west oriented transects perpendicular to the cliff outlines. Both transects showed a ~20° northwards shift in aspect during the study period, and the first transect showed a 10° reduction of the average slope which corresponded to the expansion of the cliff on upper shallower slopes followed by a progressive reburial at the base. The melt at this cliff was higher than at Cliff 1. It increased from 3.0 +/- 0.7 (3.2 +/- 1.0) cm.day$^{-1}$ at the end of May to more than double - 6.8 +/- 1.7 (6.5 +/- 0.8) cm.day$^{-1}$ - at the end of July for transect 1 (transect 2). These values are
substantially higher than the melt predicted by the energy balance model and mostly due to the **higher measured melt on the upper, shallower section of the cliff** (Fig. S5). However, for both transects there was a sharp reduction in melt synchronous with the progressive reburial of the cliff (Fig. 11). Melt then plateaued around 2.7 +/- 2.7 cm.day$^{-1}$ for transect 1 and 3.3 +/- 1.3 cm.day$^{-1}$ for transect 2 and finally decreased to almost null values in October. The modeled cliff energy balance was very similar for both transects and almost identical to that of the first cliff and while it represented the general melt patterns well,
the amplitudes were lower.

### 4.2.3 Langtang Cliff 3

Langtang Cliff 3 was a relatively large cliff (20-30 m tall, 70-100 m wide) which was predominantly north-east facing at the start of the period but expanded to north-facing slopes during the study period (Fig. 12). There was a large pond at the base of the cliff for the whole study period, which persisted throughout the season but also slowly drained by a total of 1.7 m according





to the pressure transducer record (Fig. 13c), leaving a notch at the base (Fig. 12a-f). Most of the drainage occurred in July. The north-facing debris-covered slope to the East of the cliff **started slumping at the end of July, when the notch appeared under the debris, revealing that the pond had also been undercutting this slope**. The slumping accelerated and the slope was mostly debris-free by the end of August, leading to a doubling of the cliff area between the end of July and the end of September (Fig. 13b). Most of the backwasting occurred at the shallower section of the cliff that was disconnected from the pond until the end of July, while the steeper section of the cliff in contact with the pond displayed lower backwasting rates.

All transects showed similar melt patterns. Melt for transect 1 was lower at the start of the period than for the other transects by 1.7-1.9 cm.day$^{-1}$, which was also the case at the time of peak melt in early July. Melt then decreased from early July to October, when sub-zero temperatures and snowfalls became predominant. Despite a decrease in shortwave radiation at the end of August for transect 1, **melt increased in August synchronously with a reduction in slope linked with the slumping of the North-facing slope** (Fig. S8). Transect 1 also displayed an increase in slope by 10° in June and July, prior to the cliff expansion, while the slope and aspect of transects 2 and 3 stayed consistent throughout the season, with particularly high slopes, steeper than 50° and up to 65°.

### 4.2.4 24K cliff

On 24K we focused on a set of linked cliffs at the center of the survey domain, 130 m wide and 10-20 m tall (Fig. 14, 15). These cliffs, which could also be regarded as one single cliff split by patches of thin debris, **occupied the slopes of the outer bend of a supraglacial stream, which was flowing directly at the base of the ice cliffs**, sometimes undercutting the ice slopes. The center of the bend was steeper and was occupied by a large continuous cliff, while the sides displayed a **changing combination of debris patches and bare ice** (Fig. 14). This configuration remained throughout the study period except on the west side of the meander (transect 1), where the **stream disconnected from the ice cliff during the study period** (Fig. 15a), **causing a progressive reburial of this outer section in July and August** (Fig. 14). This was just a small portion of the cliff and overall the cliff area did not change by more than 10% over the whole study period, and the aspect and slope of the different transects remained consistent (Fig. S10).

The different transects displayed comparable temporal melt and energy balance patterns (Fig. 7), with an increase from June to the mid-August peak followed by a steeper decrease until the end of the study period in early October, characterized by close to zero melt values and regular snow falls (Fig. 7b). The melt variability was driven by net shortwave radiation, which also represented more than 50% of the energy budget during the whole study period. Contrary to Langtang, the net longwave contributed positively to the cliff energy balance due to higher air temperatures and therefore higher incoming longwave from the atmosphere (Fig. 7). Transects 1, 3 and 4 displayed high (7-8 cm.day$^{-1}$) melt values at the very start of the study period, which were not represented by the energy balance model. **These high values exceeding modeled melt at the start and at peak melt, as well as the general patterns throughout the season also followed an inverse pattern with cliff brightness**





(Fig. 15c), **which was itself correlated with precipitation** (Fig. 15b). **Transects 2 and 4 were the ones where measured and modeled melt disagreed the most** (up to 25% difference), with generally lower measured melt rates than predicted by
the model, and **also the ones with the most patchy debris**, where outlining the cliff extents was particularly difficult.

## 5 Discussion

The sub-seasonal observations of ice cliff melt, evolution and the underlying processes were made possible thanks to the use of time-lapse photogrammetry, which enabled the semi-automated production of weekly DEMs of the survey area with an estimated uncertainty of +/- 20 cm (+/- 6 cm) for Langtang (24K). This is a novel approach, the advantages and drawbacks of
which we discuss in detail in the supplementary Section S1 (Use of time-lapse photogrammetry approach). Here we discuss instead the main findings that the new setup allowed, in terms of understanding cliff melt rates and contribution to mass losses (Section 7.1), as well as the processes that control their evolution and are not yet included in current models (Section 7.2).

### 5.1 Controls on ice cliff melt variability

The studied cliffs displayed melt rates at 3-9 times higher than the surrounding debris-covered ice on 24K and 6-38 times
higher on Langtang, where thick debris (>0.5 m) in the lower portion of the glacier prevents almost any sub-debris melt (Miles et al., 2021; McCarthy et al., 2021), thus promoting the melt-generating role of cliffs. While the cliff melt values are comparable to previous estimates for other debris-covered glaciers based on cliff volume loss and backwasting rates (e.g. Sakai et al., 1998; Juen et al., 2014; Brun et al., 2016; Mishra et al., 2021), the high temporal resolution estimates are more accurate, as they allow calculations of melt over strongly varying cliff geometries. Changes in cliff melt rates over time from the time-
lapse DEMs ranged between 0 and 8 cm.day$^{-1}$, and captured the progressive changes in cliff area and shape (Fig. 8-15), thus enabling a new, more precise estimate of cliff melt compared to the values extracted from the beginning and end of season DEMs (Pléiades and UAV DEMs) (Table S4). This is the first time that the sub-seasonal variability of ice cliff melt has been quantified, and it shows that use of only beginning and end of season cliffs' geometries, neglecting the history of area and geometry changes over a melt season, can lead to an underestimation of about 50% in melt rates (Table S4).

Exchange of energy with the atmosphere controlled cliffs' evolution at both sites. Cliff melt rates varied substantially over the melt season and displayed similar patterns for all cliffs, with an overall trend of increase, peaking and then decline, on which a smaller-order variability was superimposed, controlled by snow and liquid precipitation. Cliff melt variability was driven by the combination of short- and longwave radiation, and turbulent fluxes which contributed considerably during the monsoon
on Langtang, when incoming shortwave was reduced (Buri et al., 2016a). Differences in air temperature at the two sites led to a general negative net longwave radiative flux on Langtang and positive on 24K. This demonstrates the need to account for the whole energy balance to estimate cliff melt.



The ice cliffs of the two study sites were generally north-facing and received little direct illumination during the study period.
As a result, the aspect controls on ice cliff evolution described in previous studies (Sakai et al., 1998; Buri and Pelliciotti, 2018) were not evident when analyzing the ice cliff patterns (Fig. S3), and contrary to previous observations and model tests, no evidence of melt gradient was visible at the cliff surface (Fig. S4, 5, 7, 9; Buri et al., 2016a; Watson et al., 2017b).

At both sites, the measured melt was more spatially variable than predicted by the model (Fig. S4, 5, 7, 9), which used a
constant albedo and therefore was not able to account for the influence of debris on the cliff energy-balance (Table 2). Two effects were visible:

- **Melt reduction from patchy debris**: lower measured melt values evident at the foot of the Langtang Cliffs 2 and 3 (Fig. S5 d-f, S7 a-c) were likely caused by the active reburial of these sections of the cliffs during shorter time intervals
than the 2-3 week period over which melt was integrated. This influence of debris was also visible on the 24K cliff where the two transects which had the higher proportion of 'dirty' ice, and where it was most difficult to outline the ice cliff relative to the patchy debris, experienced reduced melt. At this location, the debris on the ice cliff was thick enough to reduce melt (Fig. 14-15, S9).

- **Melt enhancement from thin dust layers**: The higher melt values on the upper and shallower cliff slopes that had
recently become free of debris of Langtang Cliffs 2-3 (Fig. S5 d-f, S7 a-f) were likely caused by lower albedo values due a higher concentration of dust particles at the surface (Fyffe et al., 2020). Similarly, transect 3 of the 24K cliff was affected by small debris clasts and thin debris (Fig. 14), but these did not reduce melt and more likely led to higher melt rates due to lower albedo values.

This effect of thin debris dust on albedo was particularly visible on 24K, where cliff brightness, which we considered as a proxy for albedo, followed an inverse pattern to that of cliff melt, and was therefore likely responsible for some of the observed differences between the measured and modeled melt. Indeed, a sensitivity test conducted for transect 3 on 24K showed that a 0.1 change in cliff albedo led to a 5-10% change in melt (Fig. S11). Lower albedo values from this surface dust, unaccounted for in the model, could therefore partly explain measured melt rates 20-40% higher than predicted by the model at the start of
the study period and at peak melt on 24K. Interestingly, for the 24K cliff, changes to cliff brightness seemed to be controlled by liquid precipitation, which promoted the 'washing' of the small debris clasts that accumulated at the surface of the cliffs, thereby removing the thin surface dust layer and increasing the albedo (Fig. 14, Fyffe et al., 2020). This effect was not visible on the darker, steeper and drier cliffs of Langtang, but for glaciers like 24K it could lead to a decrease in cliff melt with the increase in occurrence of wet precipitation events at high elevation (Jouberton et al., 2022).


On the contrary, snow events at both sites in the pre- and post-monsoon periods likely reduced melt at the cliff surface. Indeed, while the snow on the debris surface usually melted away within hours after the snowfall, these north-facing steep ice slopes





had the tendency to retain the snow much longer (Fig. 8f), and up to a full day on Langtang, thereby increasing the cliff albedo and interrupting ice melt until all the snow had melted. Such effects were also not represented in the model.

**5.2 Controls on ice cliff area variability**

One of the main results of this work was that debris local dynamics are a key influence on cliffs evolution. Debris accumulating at the surface of the cliffs influenced melt, reducing it when enough patches of debris clasts had accumulated at the surface but also darkening the cliff, therefore reducing its albedo. Debris also had an influence on cliff area and slope (Table 2, Fig. 16). Debris is constantly moving on the debris-covered surface and this motion is enhanced by slope, liquid precipitation, moisture content but also debris evacuation at the base of a slope (Nicholson et al., 2018; Moore, 2018; Fyffe et al., 2020; Westoby et al., 2020). Additional debris redistribution during the wet monsoon season has even been hypothesized to increase the cliff relative area at the glacier scale (Steiner et al., 2019). While our dataset did not encompass enough cliffs to test this hypothesis, we observed considerable debris motion and areal changes at all the studied cliffs (Fig. 11b, 13b, Table 2). The planimetric area of the 24K cliff and the Langtang Cliff 1 did not change by more than 20% during the study period despite evidence of cliff lateral expansion and reburial, but Langtang Cliffs 2 and 3 experienced dramatic expansion and reburial, leading to doubling in size of Cliff 2 within the course of a month and a reduction to its initial size one month later (Fig. 11b, Table 2). Langtang Cliff 3 also underwent a 100% areal increase in 2 months (Fig. 13b). These changes demonstrate the strong temporal variability of ice cliffs at the sub/seasonal scale, which underlines the interannual dynamics of ice cliff population at the glacier scale (Steiner et al., 2019; Kneib et al., 2021).

**Debris evacuation at the base of the slope was the main controlling factor of all the cliff area change events**. For Langtang Cliffs 2 and 3, the presence of a pond and its undercutting of the cliff base, led to the instability of shallow debris-covered slopes in the vicinity of the cliff, which sustained debris evacuation from the lower portion of the cliff. On the contrary, when the pond drained at the base of Cliff 2, and thermo-erosional melt and instability of the cliff base ceased, this led to rapid reburial of the lower portion of the cliff. The same events were visible at Cliff 1 where a cavity at the cliff base led to debris evacuation, while its absence prevented any lateral cliff expansion. For the 24K cliff, partial cliff reburial was triggered by the disconnection of the supraglacial stream from the base of the cliff, which effectively 'switched off' this sediment evacuation pathway. For the remainder of the cliff, a connection between the cliff and the stream served to maintain and sometimes steepen the slope. Undercuts at the base of ice cliffs are indeed common even without the presence of a pond (Miles et al., 2016; Röhl, 2006; Kraaijenbrink et al., 2016; Watson et al., 2017b), and streams and crevassing have been shown to promote ice cliff development by serving as mechanism for the removal of debris at their base (Mölg et al., 2020; Westoby et al., 2020; Mishra et al., 2021).



### 5.3 Avenues for future research

This study confirmed the robustness of the Buri et al. (2016a) ice cliff energy balance model to derive cliff melt estimates for
a given slope and aspect over a period of 2-3 weeks. The model, which was mostly developed and evaluated using data obtained
from the Langtang catchment (Steiner et al., 2015; Buri et al., 2016a), performed well when applied to 24K, a glacier located
in a very different climatic setting (Fugger et al., 2022). This confirmed the suitability of the model to explore the melt
contribution of ice cliffs at the large scale. However, beyond a period of one month, the variability in cliff area may lead to
considerable changes in cliff extents, aspect and slope, and thus a need to better account for these aspects of cliff evolution in
the model, even in a simplistic way (Buri et al., 2016b, 2021). Future model developments in this direction should attempt to
reconcile mechanisms of cliff backwasting that are driven primarily by the cliff energy balance with debris redistribution
processes and the influence of supraglacial hydrology. Models of debris redistribution exist and have been applied to
understand the evolution of debris thickness patterns on debris-covered glaciers (Moore, 2021, 2018; Nicholson et al., 2018;
Westoby et al., 2020). Their integration into sub-debris and ice cliff melt models, along with the representation of the influence
of streams and ponds, would represent a key improvement in the numerical representation of the long-term patterns of debris-
covered glacier surface evolution and melt (Bartlett et al., 2020; Ferguson and Vieli, 2021). Furthermore, the cliff energy
balance model would also benefit from better constraints of the characteristics and temporal variability of key parameters such
as debris and ice cliff emissivity and albedo (Fig. S11), as well as a more robust interpolation of wind from the AWS to the
cliff surface (Bonekamp et al., 2020). Indeed, a major uncertainty of the cliff energy balance model outlined in previous studies
comes from the turbulent fluxes (Steiner et al., 2015) which are notoriously difficult to constrain on debris-covered glaciers
(Miles et al., 2017a; Steiner et al., 2018).

This study has highlighted the variability in ice cliff characteristics and behaviors, and has demonstrated a need for the
continued collection of high spatio-temporal resolution data of ice cliff complexes (including south-facing cliffs) and their
surroundings (varying debris thicknesses) in order to better constrain the influence of debris and supraglacial hydrology on ice
cliff evolution in different climatic and glaciological settings. Such observations, along with the identification of debris cover
characteristics (thickness, water content, displacement) at similarly high resolution would help to improve the numerical
representation of debris cover-related processes to predict the evolution of the surface topography of debris-covered glaciers
(Moore, 2021). Future surveys could be conducted with high-resolution time-lapse photogrammetry, which has proved highly
effective in this study for monitoring and quantifying surface changes of debris-covered glaciers at high temporal resolution.

### 6 Conclusions

This study bridged a crucial gap in ice cliff observations by using terrestrial time-lapse photogrammetry to quantify the sub-
seasonal evolution of four ice cliffs on two climatically contrasting Himalayan debris-covered glaciers. Prior to our work, cliffs





had been observed only at the beginning and end of the melt season (because of logistical and field challenges), but never

during this period, when most of the ablation occurs. Our main findings are as follows:

- The time-lapse camera DEMs enabled a precise quantification of the cliff melt by accounting for sub-seasonal cliff geometry changes, which are ignored when extracting melt from pre- and post-monsoon DEMs, leading to an underestimation of ice cliff melt rates.

- Sub-seasonal variability in cliff melt was high, and was driven mainly by shortwave radiation, while air temperature
was the determining factor for the sign of the net longwave contribution. Overall, the modeled melt agreed with the observations.

- The interaction of the cliffs with surrounding debris cover was found to be particularly crucial, and increased the spatial variability of the cliff melt by causing very strong changes in the cliff geometry. At the cliff surface, it had two main effects:
o  The presence of small clasts or thin layers of dust reduced the cliff albedo (resulting in increased melt).
           o  The presence of slightly thicker, often patchy debris at the cliff surface and the active reburial of parts of the cliffs reduced melt via the debris insulating effect.

- Liquid precipitation events were effective at 'washing' thin debris cover from the cliff surface and increasing its albedo, whilst snow events had a similar effect.

- Ultimately, our results confirmed that the connectivity between ice cliffs and supraglacial hydrology (streams, ponds) exerts an important control on rates and patterns of cliff expansion and reburial, and that the relevant processes and feedbacks need to be accounted for in contemporary ice cliff energy balance models to better constrain cliff melt and the long-term surface evolution of debris-covered glaciers.

**Code availability**

The Python scripts to automate the processing of the time-lapse images to DEMs, and the R scripts to calculate melt will be made available on Zenodo and GitHub upon acceptance of the manuscript.

**Data availability**

DEMs, orthoimages and melt rasters will be made available on Zenodo upon acceptance of the manuscript.

**Author contribution**

MK, ESM and FP designed the study. MK, ESM designed the time-lapse setup. MK, ESM, SF, MM conducted the fieldwork on Langtang and MK, ESM, SF, TES, ZC, MT, MJW, WY on 24K. PB provided the cliff energy-balance model and helped



in applying it. ESM provided the codes to flow-correct the different DEMs and helped in applying them. MK completed all the analysis and composed the manuscript. FP supervised the study. All authors aided in the reviewing and editing of the manuscript.

**Competing interests**

The authors declare that they have no conflict of interest.

**Acknowledgements**

This project has received funding from the European Research Council (ERC) under the European Union's Horizon 2020 research and innovation programme grant agreement No 772751, RAVEN, "Rapid mass losses of debris covered glaciers in
High Mountain Asia". F. Pellicciotti, W. Yang and M. Westoby acknowledge support from The Royal Society via a Newton Advanced Fellowship award (NA170325). The Langtang Pléiades images were acquired by CNES's ISIS programme facilitating science access to imagery. We would like to thank Simone Jola, Stefan Boss and Marco Collet at the Swiss Federal Research Institutes SLF and WSL for their help with the assembling and the testing of the time-lapse camera setups. The weather-proof box and intervalometer were purchased from Bixion, whose team provided very helpful advice for the
installation and wiring of the time-lapse cameras. Last but not least, the fieldwork was supported and organized by the Himalayan Research Expeditions (HRE) team in Nepal, and Tibetan helpers Labadunzhu and Wangqingduojia in Tibet, who made the instrument installation and data collection possible.

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

## Figures & Tables

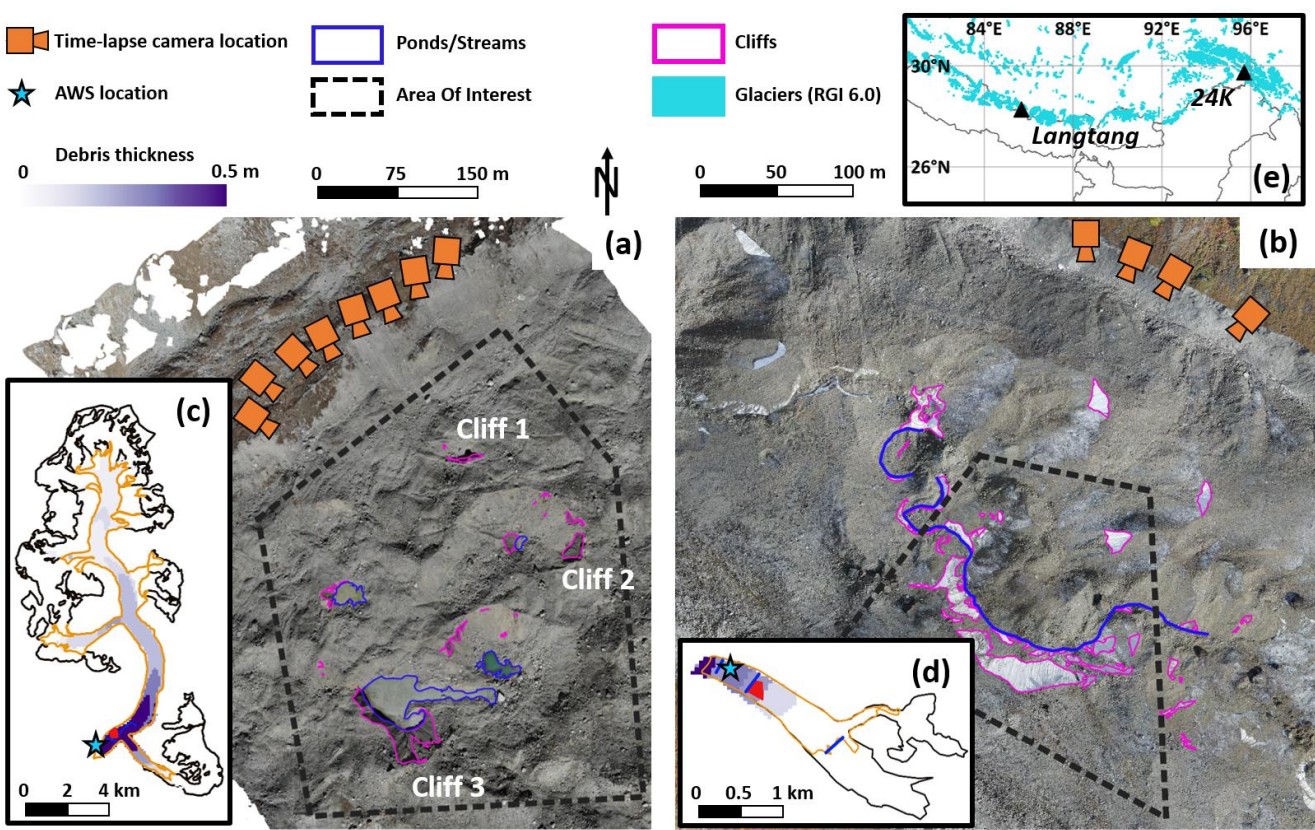

**Figure 1: Survey domains of Langtang (a) and 24K (b) Glaciers. Background is the UAV orthoimage from 02/11/2019 (Langtang) and 11/10/2019 (24K), with the outlines of cliffs (pink), ponds and streams (blue). (c-d) Glacier (black) and debris-cover (orange) outlines of Langtang and 24K with the location of the areas of interest (AOIs)(red), automatic weather stations (AWSs) and ground penetrating radar (GPR) measurements (blue). Background shows the distributed debris thickness from McCarthy et al. (2021). (e) Location of the two sites in the Himalaya.**






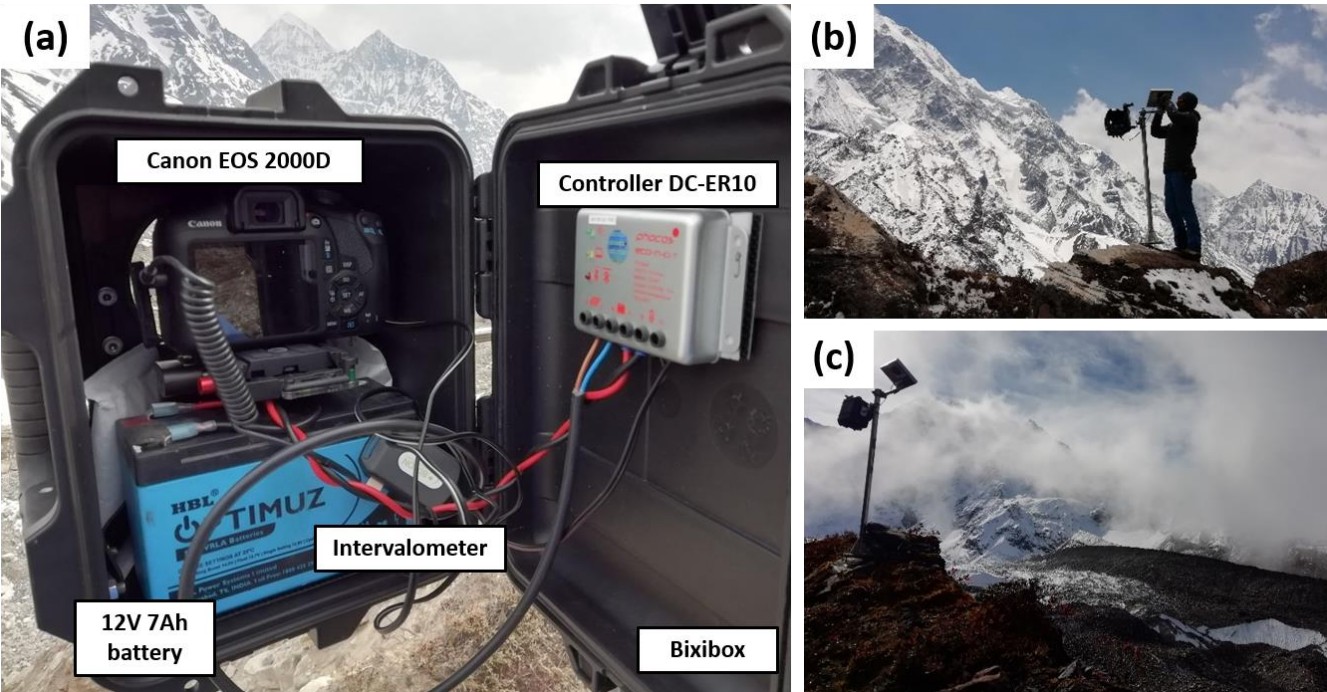

**Figure 2: (a) Different elements of the time-lapse camera setup inside the Bixibox. (b) Mounting of a time-lapse camera on the moraine of Langtang Glacier. (c) Time-lapse camera overlooking the ice cliffs of the 24K study domain.**

**Table 1: (a) Different elements of the time-lapse camera setup inside the Bixibox. (b) Mounting of a time-lapse camera on the moraine of Langtang Glacier. (c) Time-lapse camera overlooking the ice cliffs of the 24K study domain.**

| Platform | Model | Site | Date | Orthoimage & DEM Resolution (m) |
|---|---|---|---|---|
| UAV quadcopter | Mavic 2 Enterprise | 24K | 10/06/2019 | 0.12 |
| UAV fixed-wing | eBee PLUS | 24K | 11/10/2019 | 0.12 |
| UAV quadcopter | Mavic 2 Enterprise | Langtang | 02/11/2019 | 0.2 |
| Satellite stereo | Pléiades 1A | Langtang | 14/06/2019 | 2 |
| Satellite stereo | Pléiades 1A | Langtang | 22/10/2019 | 2 |




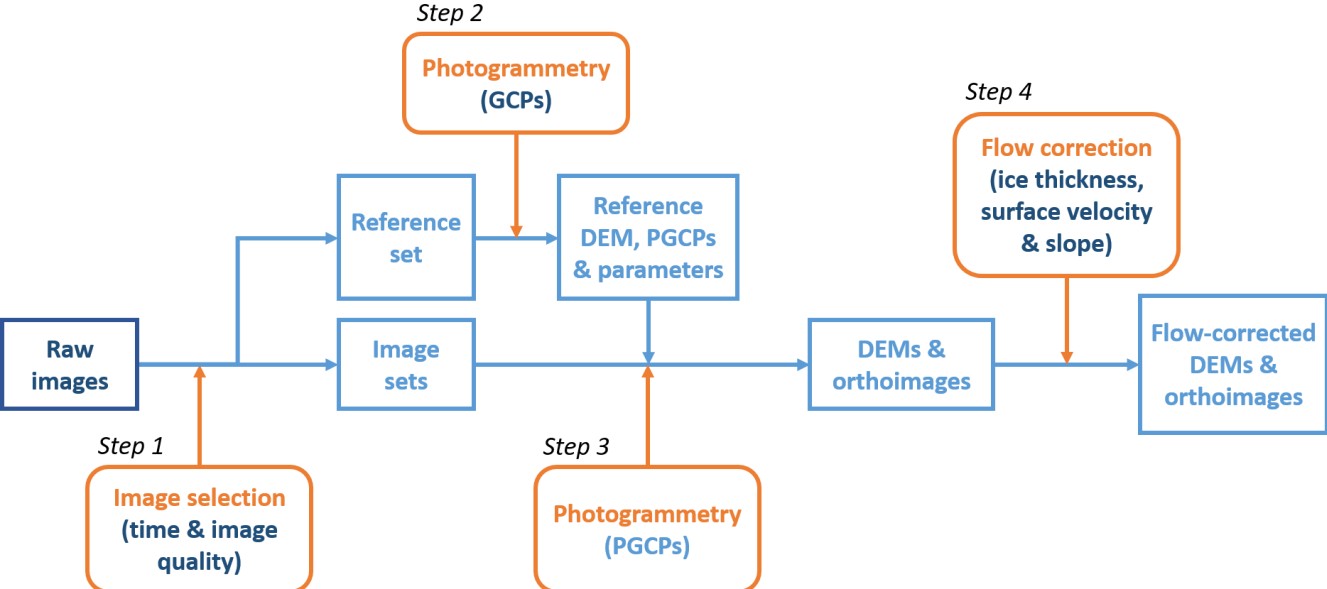

**Figure 3: Processing workflow of the time-lapse images, indicating the initial data (dark blue), processing steps (orange), intermediate and final outputs (light blue).**



**Figure 4: Survey areas with Pseudo GCPs (red), on-glacier GCPs used for the reference image sets (yellow), stream, ponds and cliff outlines, from the perspective of the orthoimages (a-b) and the reference time-lapse images (c-d), on Langtang (a & c) and 24K (b & d).**

**Uncertainties as a function of time**     **Uncertainties as a function of distance**

**Figure 5: (a-b)** dH patterns of all considered time-lapse DEMs relative to the reference DEM assessed over stable terrain as a function of time. The dotted lines indicate the DEMs furthest away in time, considered to test the uncertainty as a function of distance. **(c-d)** dH patterns relative to the DEM furthest away in time from the reference DEM and four (Langtang) and seven (24K) DEMs less than four days away, as a function of distance.





**Figure 6: (a) Elevation change from the Langtang flow corrected Pléiades DEMs (22/10/2019 - 14/06/2019). (b) Elevation change from the 24K flow corrected UAV DEMs (11/10/2019 - 10/06/2019). (c) Average measured and modeled melt from the time-lapse camera data as a function of the average slope from the time-lapse DEMs over the full study period for all blue transects in (a) and (b). The bars indicate the uncertainty of the measured melt rates.**
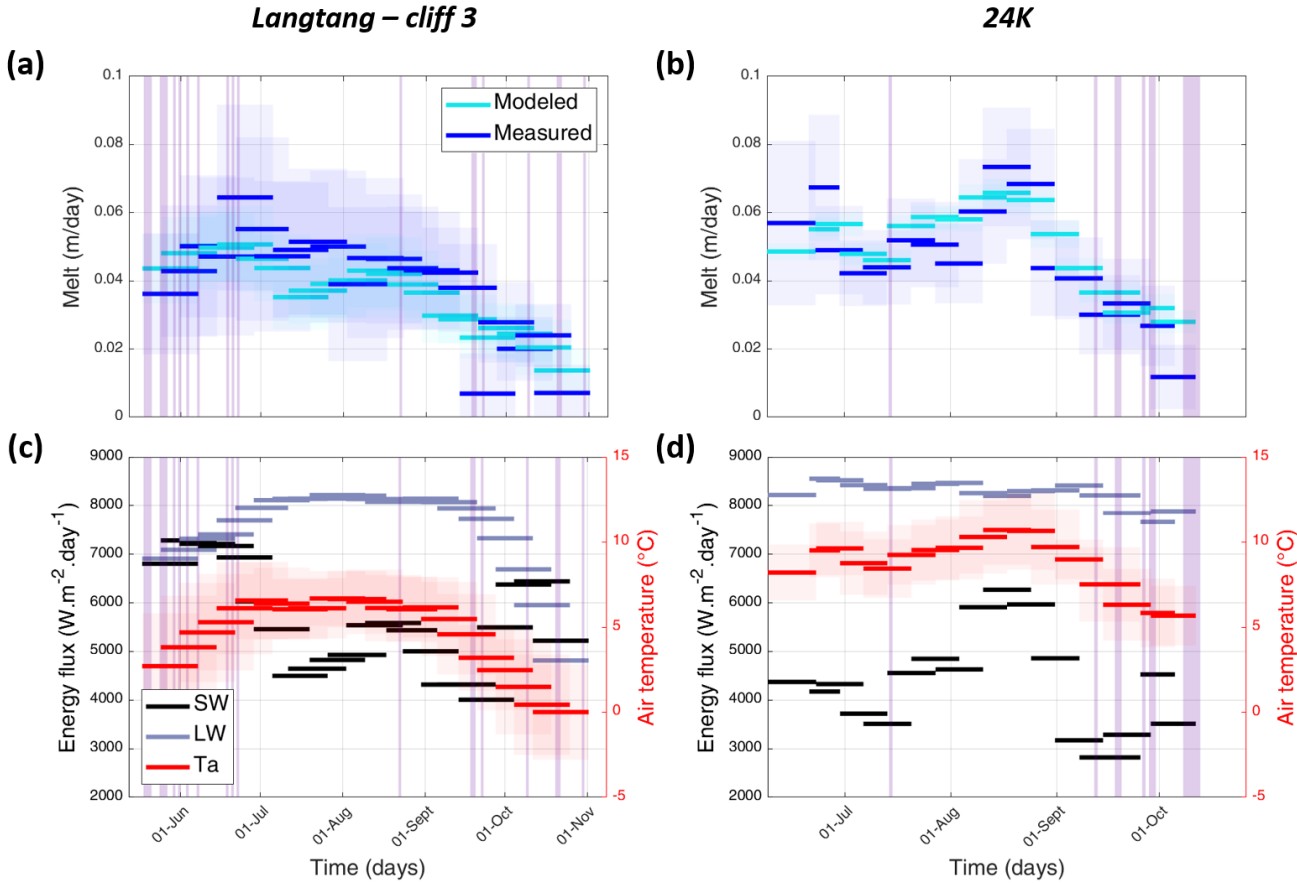

**Figure 7: Observed sub-seasonal measured and modeled melt patterns of Cliff 3 on Langtang (a) and of the 24K cliff (b). The lines show the spatially-averaged cliff melt over the different periods and the shaded areas represent the standard deviation. (c-d) Average daily incoming shortwave and longwave radiations and mean and standard deviation of air temperature over the same time periods at the AWS locations. The purple bars show the days with snow events.**



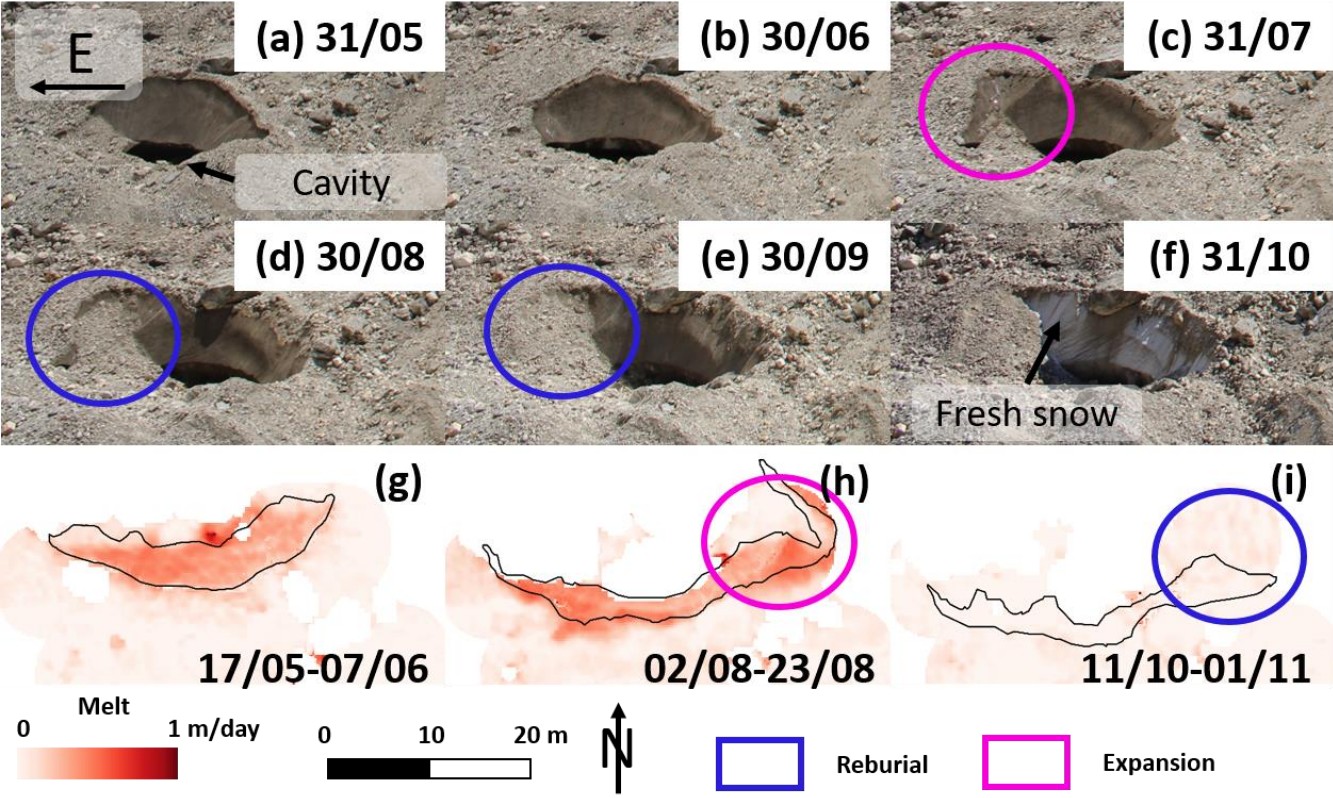

Figure 8: (a-f) Evolution of Langtang Cliff 1 throughout the study period from the time-lapse camera images. The pictures were all taken at the same time of day (11:45). The black horizontal arrow in (a) indicates east. (g-i) Tri-weekly observed melt patterns at the start, in the middle and at the end of the study period. Cliff outlines are shown in black.




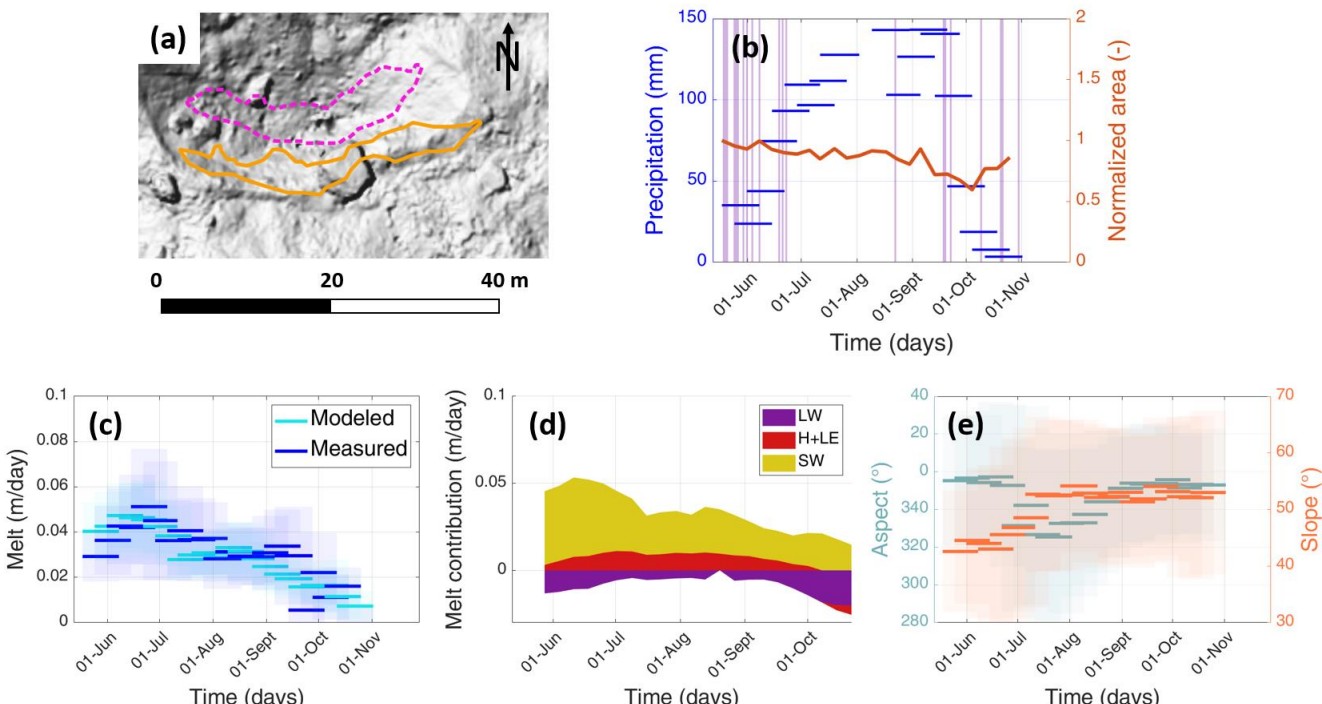

**Figure 9: (a) Langtang Cliff 1 at the start (pink dashed lines) and the end (orange full lines) of the study period. Background is the**
**hillshade of the 02/11/2019 UAV DEM. (b) Total precipitation over each time-lapse period and cliff planimetric area evolution normalized by the initial cliff area. The purple shaded areas correspond to days with snow events. (c) Observed and modeled melt as a function of time. The lines show the average value over the different periods and the shaded areas the standard deviation. (d) Modeled net energy fluxes. (e) Measured slope and aspect as a function of time.**



Figure 10: (a-f) Evolution of Langtang Cliff 2 throughout the study period from the time-lapse camera images. The pictures were all taken at the same time of day (11:45). The black horizontal arrow in (a) indicates east. (g-i) Tri-weekly observed melt patterns at the start, in the middle and at the end of the study period. Cliff outlines are shown in black and the focus transects in light blue.



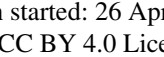



Figure 11: (a) Langtang Cliff 2 at the start (pink dashed lines) and the end (orange full lines) of the study period. Background is the
hillshade of the 02/11/2019 UAV DEM. The light blue rectangles are the cliffs' two main study transects. (b) Total precipitation over
each time-lapse period and cliff planimetric area evolution normalized by the initial cliff area. The purple shaded areas correspond
to days with snow events. (c) Pond surface temperature. (d-i) For each transect, the measured and modeled melt (d-e), modeled net
energy fluxes (f-g), and measured slope and aspect (h-i) as a function of time.





**Figure 12:** (a-f) Evolution of Langtang Cliff 3 throughout the study period from the time-lapse camera images. The pictures were all taken at the same time of day (11:45). The black horizontal arrow in (d) indicates east. (g-i) Tri-weekly observed melt patterns at the start, in the middle and at the end of the study period. Cliff outlines are shown in black and the focus transects in light blue.
**Figure 13:** (a) Langtang Cliff 3 at the start (pink dashed lines) and the end (orange full lines) of the study period. Background is the hillshade of the 02/11/2019 UAV DEM. The light blue rectangles are the cliffs' three main study transects. (b) Total precipitation over each time-lapse period and cliff planimetric area evolution normalized by the initial cliff area. The purple shaded areas correspond to days with snow events. (c) Pond water level. (d-l) For each transect, the measured and modeled melt (d-f), modeled net energy fluxes (g-i), and measured slope and aspect (j-l) as a function of time.





**Figure 14: (a-e)** Evolution of the 24K cliff throughout the study period from the time-lapse camera images. The pictures were all taken at the same time of day (10:00). The black horizontal arrow in (a) indicates east. **(f-h)** Bi-weekly observed melt patterns at the start, in the middle and at the end of the study period. Cliff outlines are shown in black and the focus transects in light blue.





**Figure 15: (a) 24K cliff at the start (dashed lines) and the end (full lines) of the study period. Background is the hillshade of the 11/10/2019 UAV DEM. The light blue rectangles are the cliffs' four main study transects on which we focus. (b) Total precipitation over each time-lapse period and cliff planimetric area evolution, normalized by the initial cliff area. The purple shaded areas correspond to days with snow events. (c) Cliff brightness. (d-o) For each transect, observed and modeled melt (d-g), modeled net energy fluxes (h-k) and measured slope and aspect (l-o) as a function of time during the full study period.**

**Table 2: Behaviors of the different cliffs studied and their controlling factors.**

| Cliff | Behavior | Controlling factors | |
|---|---|---|---|
| | | **From surrounding topography** | **At the surface of the cliff** |
| **Langtang Cliff 1** | Expansion followed by reburial | Cavity at the cliff base | |
| **Langtang Cliff 2** | • Expansion followed by reburial<br>• Heterogeneous melt patterns | • Draining pond at the cliff base<br>• Cliff backwasting and propagation to upper slopes | • Deposition of dust (low angle sections)<br>• Patchy debris |
| **Langtang Cliff 3** | • Expansion<br>• Heterogeneous melt patterns | Pond incision | • Deposition of dust (low angle sections)<br>• Patchy debris |
| **24K cliff** | • Central portion maintained<br>• Partial reburial at the edge<br>• Heterogeneous melt patterns | Stream incision | • Deposition of dust and patchy debris (low angle sections)<br>• Cliff 'washing' effect |




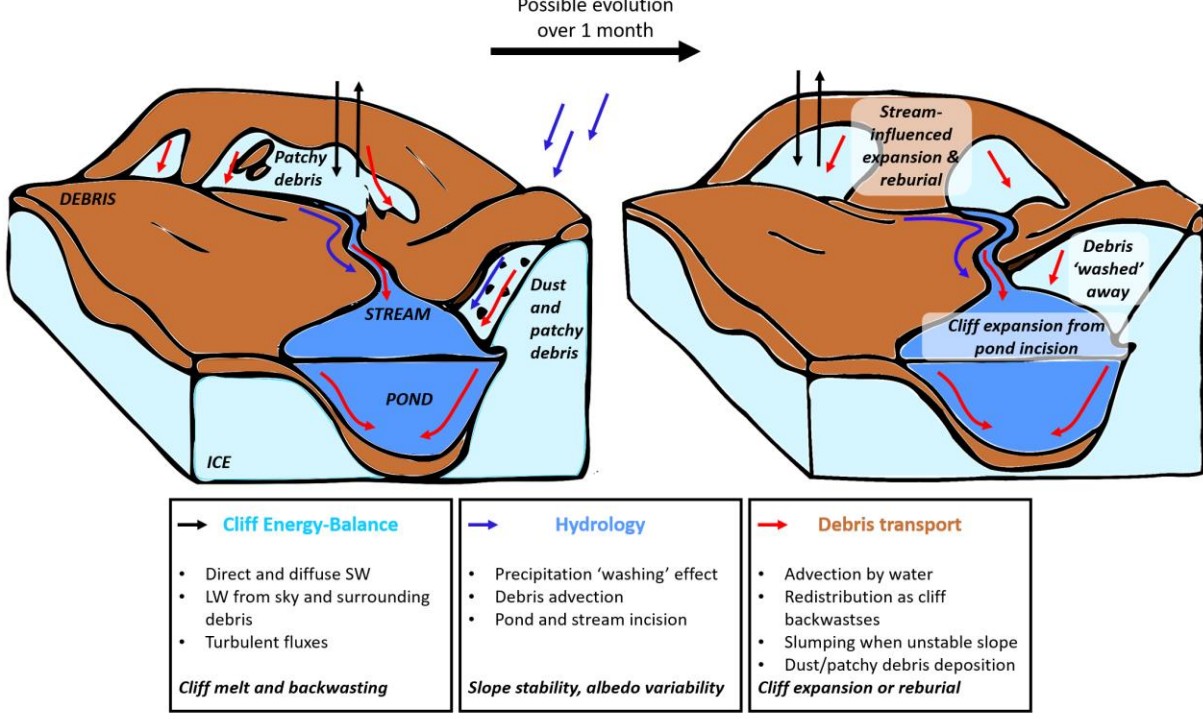

**Figure 16: Interactions between cliff energy balance, hydrology and debris transport at the surface of a debris-covered glacier highlighted by the time-lapse observations.**
