# Peer review of "Sub-seasonal variability of supraglacial ice cliff melt rates and associated processes from time-lapse photogrammetry"

_The Cryosphere, 2022_

## Author Comment (AC1)

*The reviewer's comments are in black and our answers in blue and italics.*

In this study, the authors monitored the evolution of parts of the debris covered tongue of two glaciers (Langtang Glacier and 24K Glacier) during the monsoon season of the year 2019. They developed a very innovative setup to track the 3D surface changes of the surface at a weekly resolution. They use these weekly digital elevation models (DEMs), and some additional knowledge on the ice dynamics, to calculate the melt occurring from ice cliffs at very high spatial and temporal resolution. They compare their observed cliff melt rates to model simulations to decipher the controls on ice cliff melt and evolution.

I commend the authors for the impressive amount of work behind this manuscript. I think it is a very good addition to the literature, but I have some semi-major comments, and a number of specific ones. Addressing them might help clarifying some aspects of the manuscript.

*We would like to thank Reviewer 1 for their very relevant and constructive comments. We agree that some clarification may be necessary for the points raised and we will try our best to address these concerns.*

*Specifically, we will:*

1) *Reformulate paragraph 3.5 to make the uncertainty calculation clearer and focus on the systematic uncertainties. We will also add a table summarizing the final uncertainty values.*
2) *Add explanations to the text as to why we did not use the dynamic ice cliff model and prepare additional model runs to show the comparison with the current implementation.*
3) *Streamline the manuscript by shifting some of the methods and results sections to the supplementary material.*

Major comments:

The uncertainty assessment (section 3.5) is generally done in a careful way. The authors did their best to evaluate honestly the uncertainty associated with their newly developed technique. However, I have some concerns with equation 4, which basically assumes that the errors are uncorrelated (independent) for each pixel. In this study, the errors should be largely correlated because they can originate from e.g. a non-perfect adjustment of the camera, or from the ice flow correction. I therefore suggest to revise this equation, and the text afterwards (L344-346). It is also not so clear, whether these uncertainties are re-used after in the text and figures, because for the figures, they refer to the standard deviation, and not to the uncertainty analysis.

*Our uncertainty analysis of the DEM differencing has shown that it could be decomposed into 1) a systematic (or correlated) error, given by the maximum absolute mean values of elevation change over stable terrain and 2) a random (or uncorrelated) error, given by the standard deviation of elevation change over stable terrain (Fig. 5). We could actually show that the bias could be considered negligible relative to the random error.*

*As pointed out by the Reviewer, Equation 4 only applies to the random error, and we agree that the current formulation of paragraph 3.5 was confusing in this sense. We will therefore reformulate it entirely to make these different points clearer and remove equation 4, as it is not particularly useful for everything that follows.*

*For simplicity, in the figures we indeed showed only the standard deviation of melt across the transects, which accounts for 1) the random error from the DEMs and 2) the melt variability at the surface of the cliff, and did not include the systematic uncertainty, which is much smaller, to avoid overloading the figures. We will make this clear in the text and will add a table showing the different components of the uncertainties.*

*Section 3.5 will therefore be rewritten as:*

[revised manuscript text omitted]

My second major comment is about the model used in the study. I don't fully understand why the authors used the static version of the model instead of the dynamic one. The dynamic model would be a great way to assess the share of each process (surface energy balance vs. 'geomorphic' processes). As it stands the manuscript is a bit frustrating, because the description of processes related to the redistribution of debris remains extremely descriptive. I am also curious about the model calibration, if there is any, because no details are provided about it.

*This is a very good question, which we are happy to clarify. Indeed, the dynamic-geometry cliff model presented in Buri et al., (2016b) was extended from the static-geometry cliff energy-balance model presented in Buri et al., (2016a) to one able to predict cliff evolution (and thus simulating changes in slope, aspect and size) by taking into account both melt and debris redistribution and neighbouring supraglacial lakes. The reviewer is correct that, ideally, a dynamic model would be appropriate for the purpose of our manuscript. However, the dynamic model was formulated and constrained with limited data of cliff geometry (a pre- and a post- ablation season DEM for each of four cliffs on one glacier). Due to the limited data available for the model development and the underlying risk of equifinality, the processes influencing the cliff dynamics (debris redistribution, additional melt from ponds) were represented by rather simple parameterisations lumping together distinct physical processes. For example, debris redistribution was simply constrained by a slope threshold above which debris would be removed, without accounting for the actual debris motion or mass conservation (Buri et al., 2016b; Moore, 2018). Similarly, the pond influence was represented by an additional melt rate at the base of the cliff, constant over the entire cliff-pond interface, and calculated from the modeling of one pond only. As such, we feel that those parameterisations and corresponding empirical parameters are too simple to represent the complexity of changes occurring during one melt season - whereas they were appropriate to provide bulk changes over long periods (the entire season, or monthly intervals, as in Buri et al., 2016b, 2018, 2021).*

*Our objective for the modelling in this study was to understand the spatiotemporal variability of the energy-balance and melt patterns of the cliffs at very high resolution. For this purpose, leveraging the actual cliff geometry at each time-step is extremely beneficial, as the actual and modelled cliff geometry will diverge even over short time-scales (1 week) due to the cliff process complexity and the lumped process representation in the dynamic model discussed above. As such, the dynamic model would be less reliable for understanding the local energy*

*balance, even if calibrated to the studied cliffs. Overall, we found that the mixed approach (observations of cliff geometry to drive the energy-balance model, which is the model component for which we have high confidence) allows to leverage the best of the observations (high resolution, frequent DEMs) and the best of the advanced model of Buri et al., 2016a (its EB component) to understand with unprecedented resolution and detail cliff melt patterns. The further development of a dynamically evolving cliff geometry model appropriate for high temporal resolution is of high interest to our group of authors and this line of research, but it would require a substantial investment to collect additional data from more cliffs, and more than two glaciers, to make the model physically representative, and it was thus outside the scope of this work - where our main goal was to understand the complex patterns of short term cliff evolution over the studied cliffs. The key step forward in our study is the ability to constrain cliff geometry changes on a weekly basis and calculate an adjusted energy balance (notably from radiative fluxes) based upon a known cliff geometry. Accordingly, we retain a high confidence that we are modelling the energy balance well at the surface.*

*We do feel that the development of a more process-oriented dynamic cliff model would be an important advance for the community. Advances in the past few years may contribute towards this, for instance improvements in the representation of debris motion at the surface of glaciers (Moore, 2018; 2021; Anderson and Anderson, 2018; van Woerkom et al., 2019; Westoby et al., 2020). While there are still some important knowledge gaps (for example related to the sliding of debris on steeper slopes or the debris evacuation by streams or ponds), we are convinced that the way forward for the cliff dynamic model would be to represent these processes in a much more physical way. This was not possible a few years ago, but the multitemporal UAV or time-lapse datasets (DEMs, orthoimages) that have been produced in recent years (Westoby et al., 2020; Sato et al., 2021; this study) should enable this next major step in future work.*

*Despite all these different elements, we understand the interest of the reviewer regarding the ability of the dynamic model to represent the melt and evolution of ice cliffs at different sites. We will therefore conduct additional tests with the cliff dynamic model to show how it compares with the current model formulation. Also, we will indicate the different points mentioned above directly in the paper by adding 1-2 sentences in Sections 3.7. and 5.3.*

*For the descriptiveness of the text regarding debris motion, we will streamline the results description in section 4.2 and focus on the main findings and move the more descriptive results to the SI.*

*Finally, to answer the question related to the calibration of the energy-balance model, we used the exact same parameters as Buri et al., (2016a), no further calibration was conducted and the same parameters were used for both glaciers. We will make sure to specify this point in the first paragraph of section 3.7.*

My third comment is about some sentences in the discussion and conclusion that I find slightly misleading, or not well supported by the data. For instance, in l253-254, l400-403 and l631-633, it is written that time and space integrated methods lead to an underestimation of melt rates of 50%. This is not really correct, it is just that the methods are looking are measuring different targets: previous methods measured integrated losses that include

reburial and expansion, and thus also melt beneath debris, while here the authors focus on shorter time scale and on melt rates along cliff transects. The same comment applies for the comparison between the melt estimates calculated for the ice cliffs vs. the sub-debris melt (e.g. L23-25 and L394-399). The calculation of this ratio also lacks details, because little is said about the uncertainties and the representativeness of each end-member.

*Indeed, our study has been focusing on data with very high spatio-temporal resolution, something that had not previously been done for ice cliffs. We believe that the comparison we made with other methods is interesting as it shows that depending on the definition adopted for the calculations, the melt contribution of ice cliffs can change considerably. The point here is to put the different approaches into perspective and show what is missed through the mixing of ice cliffs and sub-debris melt when more simple approaches are taken due to less temporarily resolved data. Similarly, a recent study has also highlighted the differences between extracting melt vertically and normal to the slope (Mishra et al., 2021).*

*Some of the statements we made may have been misleading in this sense and we will correct them as follows:*

*L400-403: 'The high temporal resolution of this dataset enables one to precisely estimate the melt contribution of ice cliffs, which is 4 to 129% greater than if the Pléiades DEMs were used for Langtang and up to 27% more than if the UAV DEMs were used for 24K (Table S4), due to the mixing of ice cliff and sub-debris melt contributions for less temporally-resolved data.'*

*L631-633: 'The time-lapse camera DEMs enabled a precise quantification of the cliff melt by accounting for sub-seasonal cliff geometry changes, which are ignored when extracting melt from pre- and post-monsoon DEMs.'*

*Regarding the comparison between the melt estimates calculated for the ice cliffs VS sub-debris melt, we feel that it is also a relevant one when thinking about ice cliff enhancement factors relative to sub-debris melt, which is a key metric to assess the contribution of ice cliffs to the melt of debris-covered glaciers (e.g. Brun et al., 2018; Mishra et al., 2021; Rounce et al., 2021). The calculation was not explicitly described in the text and we will update this and take the mean+/-STD of cliff melt and divide it by the mean+/-STD of sub-debris melt to make sure that it is representative (L394-399). L23-25 we will simply remove the comparison with the sub-debris melt rates, as it is not so relevant for the abstract.*

I found the paper slightly lengthy in some places. For instance, the introduction could be more concise, or the line 361-370 about the surface energy balance model are not extremely useful. While this is not a major issue, it would streamline the manuscript to check for each sentence whether it is relevant for the general paper scope. Similarly, the results section 4.2 that presents each cliff's evolution is difficult to follow. The authors do not link very well the morphologic and meteorological changes happening to changes in the melt rates, and instead follow a more descriptive approach.

*We thank the reviewer for their comments on this. We will make sure to streamline the sections mentioned here and make sure to outline the link between morphologic and meteorological change to changes in melt rates. Specifically we will:*

1) *Cut some of introduction sentences and especially condense paragraph L70-82*
2) *Streamline section 3.2 and stick to a shorter summary of the methods*
3) *Condense the text in section 3.5 and make it more to the point (as shown in previous comment)*
4) *Streamline the paragraph L361-370 and remove all the repeated content from the introduction*
5) *Focus on the most important results for each cliff in section 4.2 and make the links between meteorology and dynamics clearer. The remaining and more descriptive results will be moved to the Supplementary Material.*
6) *Condense L618-625 and merge with previous paragraph.*

Specific comments:

*We thank the reviewer for their thorough reading of our manuscript and will make sure that all these points are corrected as suggested.*

In some places, there are some sentences in bold font. They should be in normal font.

*We will remove the bold fonts.*

L18: "Tibet" -> should be "China" for consistency with "Nepal", no?

*Agreed.*

L20-21: the uncertainty is given at pixel of cliff scale?

*This is the systematic error of the DEMs scaled with distance calculated in Section 3.5. This should actually be divided by 2 as this is for the DEM positioning rather than the DEM differencing. This will be clarified in the text: 'We derive weekly flow-corrected DEMs of the glacier surface with a maximum bias of +/-0.05 m for Langtang Glacier and +/-0.03 m for 24K Glacier…'*

L27: I didn't find elements in the text that supports this statement in a quantitative way.

*This refers to L543-548 of the discussion. We will make sure to specify these values there.*

L36: it's the ice and not the "ice cliff" that is directly exposed. The sentence should be re-phrased.

*Agreed.*

L45: consider adding "for specific locations".

*Agreed. Will add it.*

L115: which part of the glacier section is included in the calculation of slope?

*This is the slope of the whole glacier, but since the focus is on the characteristics of the debris-covered portions of the glaciers, we will indicate the slope of the debris-covered parts instead (this will not change the rest of the sentence).*

L130: is it relevant to quote the price in a scientific publication? I suspect anyone interested by the setup will contact the authors.

*We will remove it.*

L191: maybe a slightly more quantitative assessment of the quality of the co-registration would be good

*We will indicate the improvement in terms of velocity over the off-glacier terrain.*

L193-195: the mean elevation difference on stable ground is very large (larger than the absolute value of the emergence for 24K!), and would imply a potential systematic bias. Is the median at zero? The authors should consider putting the mean or median off-glacier elevation at zero.

*This is a good point. We will make sure to put the mean to 0 and update the figures as needed.*

L292-293: why not calculating all the uncertainties within the normal intersection framework?

*We chose the conservative approach of taking the vertical uncertainty as the melt uncertainty as this removes the dependence on the slope of the off-glacier terrain and its (lack of) representativeness for the ice cliffs. We will specify this point better here.*

L330: how does the 2° uncertainty in the slope angle translates into the sigma_flow? Also the units of the different components of sigma_flow are never explicitly written.

*We will make sure to specify these elements in the text.*

L413: missing figure number

*Thanks for spotting this, we will correct it.*

L423-432: what is the impact on the melt rate?

*We will make sure to link the observations with the changes in melt rates in Section 4.2.*

L539-554: here I would expect a quantitative analysis, which remains too descriptive

*We will specify the corresponding values for these two paragraphs.*

L571: a bit in contradiction with L571

*Apologies but we do not see this contradiction?*

L603-605: then why not testing the dynamical parametrization of the model?

*We have provided a detailed answer to this point in the major comments.*

Section 5.3: it would be worth acknowledging the limitations of the study (very small area surveyed, only one full ablation season, north facing cliffs only…)

*Agreed. We will add a sentence acknowledging these limitations specifically.*

L627: "bridged a crucial gap" -> this sounds a bit like overselling the study

*We will replace it with 'considerably improved our understanding of ice cliff evolution'*

L627-648: the conclusion is mostly based on very general sentences, and does not do a good job in highlighting the specific findings of the study. It should be re-written to better highlight the novel aspects of this study. I would recommend not to use too many bullet points in the conclusion.

*We will remove the main bullet points and instead add details/numbers to highlight the specific findings.*

Fig. 7 a and b: if I read the caption well, it says that the shading area represents the standard deviation, but what standard deviation? The spatial one? Why not using the uncertainties calculated from the uncertainty analysis for the observed melt rates?

*See detailed response to the major comment.*

Fig. 7 c and d: there is a unit problem for the energy fluxes. They are not the right order of magnitude.

*Thanks for this. We will double-check these values.*

Fig. 8: is there an influence from the large boulder on top of the cliff edge? It is never mentioned in the text I think?

*We have not identified any clear effect, but will mention it in the results.*

Fig. 9 and all the following ones with the same design:

- I find it very confusing to represent the cumulative precipitation for overlapping periods, it suggests a huge amount of precipitation, because most periods are counted twice. I suggest expressing the precipitation in mm/day instead.

*We will change to mm/day.*

- On the panel d: at which time resolution are the fluxes plotted? When two model runs overlap, which one is used for the flux plot? Also fluxes should be in W m-2 and not melt contribution, because they can be negative.

*We took the central value of each period, we will specify it in the caption. We will however keep the fluxes as 'melt contribution (m/day)' to keep the link with the left panels, and specify the equivalence in the caption.*

---

## Author Comment (AC2)

*The reviewer's comments are in black and our answers in blue and italics.*

This submission presents measured and modelled melt from four ice cliffs located on two Himalayan glaciers with contrasting geomorphological characteristics. Measured melt is derived from terrestrial timelapse photogrammetry and modelled melt is derived from a surface energy balance model partly developed by the co-authors. The methods are extensively described and largely based on previous work and are sound. There is careful attention given to quantifying uncertainty and I can find no fault in that regard. The results show some interesting spatial and temporal variability in melt patterns and provide a window into the dynamics of the ablation season, which is otherwise all too often obscured by cloud and inaccessible for field observations. Interpretation of the key controls of this variability is convincing.

*We would like to thank Reviewer 2 for the thorough read of our manuscript and for their very constructive comments. Specifically, we will:*

1) *Add explanations to the text as to why we did not use the dynamic ice cliff model and prepare additional model runs to show the comparison with the current implementation.*
2) *Streamline the manuscript by shifting some of the methods and results sections to the supplementary material.*

It is notable that the premise of the work is to be able to quantify and characterise the evolution of these ice cliffs during their most dynamic period, yet the modelling that is to shed insight into the energy balance is static. There is one line of justification for this (373-374) but it could do with much better justification, especially given the most recently published work of the co-authors describes and uses the dynamic model in a similar vein and some of the future work suggested in later sections has already been realised.

*This is a very good point, which we are happy to clarify. Indeed, the dynamic-geometry cliff model presented in Buri et al., (2016b) was extended from the static-geometry cliff energy-balance model presented in Buri et al., (2016a) to one able to predict cliff evolution (and thus simulating changes in slope, aspect and size) by taking into account both melt and debris redistribution and neighbouring supraglacial lakes. The reviewer is correct that, ideally, a dynamic model would be appropriate for the purpose of our manuscript. However, the dynamic model was formulated and constrained with limited data of cliff geometry (a pre-and and a post-ablation season DEM for each of four cliffs on one glacier). Due to the limited data available for the model development and the underlying risk of equifinality, the processes influencing the cliff dynamics (debris redistribution, additional melt from ponds) were represented by rather simple parameterisations lumping together distinct physical processes. For example, debris redistribution was simply constrained by a slope threshold above which debris would be removed, without accounting for the actual debris motion or mass conservation (Buri et al., 2016b; Moore, 2018). Similarly, the pond influence was represented by an additional melt rate at the base of the cliff, constant over the entire cliff-pond interface, and calculated from the modeling of one pond only. As such, we feel that those parameterisations and corresponding empirical parameters are too simple to represent the complexity of changes occurring during one melt season - whereas they were appropriate to provide bulk changes over long periods (the entire season, or monthly intervals, as in Buri et al 2016b, 2018, 2021).*

*Our objective for the modelling in this study was to understand the spatiotemporal variability of the energy-balance and melt patterns of the cliffs at very high resolution. For this purpose, leveraging the actual cliff geometry at each time-step is extremely beneficial, as the actual and modelled cliff geometry will diverge even over short time-scales (1 week) due to the cliff process complexity and the lumped process representation in the dynamic model discussed above. As such, the dynamic model would be less reliable for understanding the local energy balance, even if calibrated to the studied cliffs. Overall, we found that the mixed approach (observations of cliff geometry to drive the energy-balance model, which is the model component for which we have high confidence) allows to leverage the best of the observations (high resolution, frequent DEMs) and the best of the advanced model of Buri et al., 2016a (its EB component) to understand with unprecedented resolution and detail cliff melt patterns. The further development of a dynamically evolving cliff geometry model appropriate for high temporal resolution is of high interest to this group of authors and this line of research, but it would require a substantial investment to collect additional data from more cliffs, and more than two glaciers, to make the model physically representative, and it was thus outside the scope of this work - where our main goal was to understand the complex patterns of short term cliff evolution over the studied cliffs. The key step forward in our study is the ability to constrain cliff geometry changes on a weekly basis and calculate an adjusted energy balance (notably from radiative fluxes) based upon a known cliff geometry. Accordingly, we retain a high confidence that we are modelling the energy balance well at the surface.*

*We do feel that the development of a more process-oriented dynamic cliff model would be an important advance for the community. Advances in the past few years may contribute towards this, for instance improvements in the representation of debris motion at the surface of glaciers (Moore, 2018; 2021; van Woerkom et al., 2019; Anderson and Anderson, 2018; Westoby et al., 2020). While there are still some important knowledge gaps (for example related to the sliding of debris on steeper slopes or the debris evacuation by streams or ponds), we are convinced that the way forward for the cliff dynamic model would be to represent these processes in a much more physical way. This was not possible a few years ago, but the multitemporal UAV or time-lapse datasets (DEMs, orthoimages) that have been produced in recent years (Westoby et al., 2020; Sato et al., 2021; this study) should enable this next major step in future work.*

*Despite all these different elements, we understand the interest of the reviewer regarding the ability of the dynamic model to represent the melt and evolution of ice cliffs at different sites. We will therefore conduct additional tests with the cliff dynamic model to show how it compares with the current model formulation. Also, we will indicate the different points mentioned above directly in the paper by adding 1-2 sentences in Sections 3.7. and 5.3.*

There is also a great deal of attention given to the important role that albedo plays in controlling melt, and the fact that it is dealt with as a constant in the modelling, but on the other hand the timelapse images are normalised to account for illumination variability and cliff brightness is derived. It strikes me it would be a small step to use these data more explicitly to drive a time-varying albedo parameterisation that would bring the modelled and measured data more closely together. I don't suggest that the authors should implement that as a revision, but maybe the concept (and the challenges that might lie therein) could be included within the discussion of future work.

*This is an excellent point that we will add to Section 5.3 of the discussion as a further avenue to explore. There are indeed already studies that have looked into estimating albedo from RGB images (e.g. Ayala et al., 2016; Burger et al., 2018).*

Lastly, the manuscript is very lengthy, and could be streamlined in places. An example is in the description of the SfM setup, which is based on previous published work - a short summary of the departures from this previous work rather than extensive description should suffice. Lines 70-99 could be distilled into a few lines. Lines 618-625 seem to offer little other than some generic thought. A re-read of the manuscript with a critical eye for what is and what is not required, and whether any material (methods mostly) could go into Supplementary, would help to keep the reader's attention and lead them to the take-home message more efficiently.

*We thank the reviewer for their comments on this. We will make sure to streamline the sections mentioned and shorten the text where needed. Specifically we will:*

1) *Cut some of introduction sentences and especially condense paragraph L70-82*
2) *Streamline section 3.2 and stick to a shorter summary of the methods*
3) *Condense the text in section 3.5 and make it more to the point.*
4) *Streamline the paragraph L361-370 and remove all the repeated content from the introduction*
5) *Focus on the most important results for each cliff in section 4.2 and make the links between meteorology and dynamics clearer. The remaining and more descriptive results will be moved to the Supplementary Material.*
6) *Condense L618-625 and merge with previous paragraph.*

The manuscript is otherwise very well written. I have picked up some small ambiguities or points for clarification that follow here:

line20: is that horizontal or vertical uncertainty?

*Vertical. We will specify it here.*

line28: variability across space or through time? On an individual cliff, or between sites?

*We will specify 'spatio-temporal variability in cliff area at each site'*

line40: I prefer not to suggest that ice cliffs enhance melt - rather it's the debris that is supressing it (unless thin of course). Maybe exceed is a better choice of word.

*This is in line with previous research on ice cliff enhancement factors (Brun et al., 2018; Miles et al., 2022). We would keep 'enhance' but will put 'relative to their surrounding debris-covered area' next to it to make sure that there is no misinterpretation of the wording.*

line43: what is an 'advanced' energy balance model in this context?

*We will remove this adjective.*

line77: are ponds a process? Maybe pond filling and drainage? Similarly for streams. Maybe you mean down-cutting?

*Yes. Will modify as suggested.*

line121: directions of the compass don't need capitalising

*We will remove the capital letters.*

line138: add 'satellite' for those not familiar with Pleiades

*Agreed.*

line164-165: this is presumably important for the modelling? Maybe state that if so?

*The model used does not account for lake effects. We will keep this section as it is.*

line190: 'identified in the June flight'

*Will change as suggested.*

line223: 'As an initial estimate, we used the values provided by...'

*Will change as suggested.*

line225: didn't change by more than five centimetres in which direction? Not sure I follow. Do you mean five degrees?

*Here we mean the height of the camera along the mast. We will state this explicitly in the text.*

line428 and elsewhere: why the need to put text in bold? The aim being in bold made sense (perhaps) but not the rest…

*We will remove the bold parts (except for the aim).*

Table 1: caption needs attention (repeats that from the figure directly above)

*Thanks for spotting this. We will make sure to change this.*

Figure 8: clarify which period the cliff outline relates to (start or end of observation period)?

*Good point. We will specify this (they correspond to the start of the period).*

Figure 16: this is nicely presented, but the integration of it into the text is poor. It also represents one possible pathway of evolution over a discrete (set) period of a month, showing two points in time. This doesn't fit well with the rest of the study that tells the reader there is great spatial and temporal variability in behaviour, and it has been characterised at fine temporal resolution for the first time. The figure either needs better explanation in the text, revising (to really show the new information gleaned from this study), or removing.

*We will expand on the explanation of the figure in the text and present it as one possible pathway of evolution. The purpose of this figure was to indicate the different mechanisms outlined in the study, which we will explicitly state in the first paragraph of the discussion.*

**References**

Ayala, A., Pellicciotti, F., MacDonell, S., McPhee, J., Vivero, S., Campos, C., Egli, P., 2016. Modelling the hydrological response of debris-free and debris-covered glaciers to present climatic conditions in the semiarid Andes of central Chile. Hydrol. Process. 30, 4036–4058. https://doi.org/10.1002/hyp.10971

Burger, F., Ayala, A., Farias, D., Thomas, |, Shaw, E., Macdonell, S., Brock, B., Mcphee, J., Pellicciotti, F., 2018. Interannual variability in glacier contribution to runoff from a high-elevation Andean catchment: understanding the role of debris cover in glacier hydrology. https://doi.org/10.1002/hyp.13354

Buri, P., Pellicciotti, F., Steiner, J.F., Miles, E.S., Immerzeel, W.W., 2016. A grid-based model of backwasting of supraglacial ice cliffs on debris-covered glaciers. Ann. Glaciol. 57, 199–211. https://doi.org/10.3189/2016AoG71A059

Buri, P., Miles, E.S., Steiner, J.F., Immerzeel, W.W., Wagnon, P., Pellicciotti, F., 2016. A physically based 3-D model of ice cliff evolution over debris-covered glaciers. J. Geophys. Res. Earth Surf. 121, 2471–2493. https://doi.org/10.1002/2016JF004039

Buri, P., Pellicciotti, F., 2018. Aspect controls the survival of ice cliffs on debris-covered glaciers. Proc. Natl. Acad. Sci. 115, 4369–4374. https://doi.org/10.1073/pnas.1713892115

Buri, P., Miles, E.S., Steiner, J.F., Ragettli, S., Pellicciotti, F., 2021. Supraglacial Ice Cliffs Can Substantially Increase the Mass Loss of Debris-Covered Glaciers. Geophys. Res. Lett. 48. https://doi.org/10.1029/2020GL092150

Mishra, N.B., Miles, E.S., Chaudhuri, G., Mainali, K.P., Mal, S., Singh, P.B., Tiruwa, B., 2021. Quantifying heterogeneous monsoonal melt on a debris-covered glacier in Nepal Himalaya using repeat uncrewed aerial system (UAS) photogrammetry. J. Glaciol. 1–17. https://doi.org/10.1017/JOG.2021.96

Moore, P.L., 2018. Stability of supraglacial debris. Earth Surf. Process. Landforms 43, 285–297. https://doi.org/10.1002/esp.4244

Moore, P.L., 2021. Numerical Simulation of Supraglacial Debris Mobility: Implications for Ablation and Landform Genesis. Front. Earth Sci. 9. https://doi.org/10.3389/feart.2021.710131

Sato, Y., Fujita, K., Inoue, H., Sunako, S., Sakai, A., Tsushima, A., Podolskiy, E.A., Kayastha, R., Kayastha, R.B., 2021. Ice Cliff Dynamics of Debris-Covered Trakarding Glacier in the Rolwaling Region, Nepal Himalaya. Front. Earth Sci. 9, 398. https://doi.org/10.3389/FEART.2021.623623/BIBTEX

Van Woerkom, T., Steiner, J.F., Kraaijenbrink, P.D.A., Miles, E.S., Immerzeel, W.W., 2019. Sediment supply from lateral moraines to a debris-covered glacier in the Himalaya. Earth Surf. Dyn. 7, 411–427. https://doi.org/10.5194/esurf-7-411-2019

Westoby, M.J., Rounce, D.R., Shaw, T.E., Fyffe, C.L., Moore, P.L., Stewart, R.L., Brock, B.W., 2020. Geomorphological evolution of a debris-covered glacier surface. Earth Surf. Process. Landforms 45, 3431–3448. https://doi.org/10.1002/esp.4973

---

## Author Response (AR1)

**Reply to comments on the manuscript**

**'Sub-seasonal variability of supraglacial ice cliff melt rates and associated processes from time-lapse photogrammetry'**

Marin Kneib, Evan S. Miles, Pascal Buri, Stefan Fugger, Michael McCarthy, Thomas E. Shaw, Zhao Chuanxi, Martin Truffer, Matthew J. Westoby, Wei Yang, and Francesca Pellicciotti

**Response to editor and general revision**

Dear authors,

Thank you for your comments to the reviewers. The two reviewers provided some suggestions to clarify uncertainty assessment, model selection, and time-varying albedo parameterisation. Additionally, they both pointed out that the paper is lengthy. I agree with these comments and encourage you to take into account these comments in the revised paper.

Best,

Dr. Kang Yang Editor, The Cryosphere

**Dear Prof. Yang,**

Many thanks for the consideration of our manuscript. We have received two wellinformed reviews which are overall very positive on the quality of the analysis and relevance of the science presented in this manuscript. Both reviewers also raised some valid points related to the uncertainty estimation, model selection and albedo parameterization.

We would like to thank very much all reviewers for their constructive and thoughtful comments, which have surely contributed to improve the manuscript. In response to these comments, we have conducted the following changes:

- **Uncertainty assessment**: We have entirely reformulated paragraph 3.5 to make the uncertainty calculation clearer. We have also added a table in the main text with the different uncertainty values.
- **Model selection**: The choice of the static model updated with the time-lapse DEMs was made because the dynamic cliff model was not meant to represent

local processes influencing the cliff evolution (debris redistribution, pond influence) at such a fine temporal scale. As the goal of this study was to use the model to isolate the energy-balance from the cliff evolution, using the dynamic model would have led to errors in the energy balance and would have made the melt patterns difficult to interpret. By using the static model with updated geometry, we ensure that the energy fluxes are as accurate as possible. Indeed, our results show that the updated geometry model predicts melt estimates in line with our observations. We have provided a very detailed response to these comments, including figures of the cliff geometry changes when the dynamic model is run at such a high spatiotemporal resolution without recalibration of the geometric change parameters.

- Albedo parameterization: We have now added some details about this topic in the discussion. In short, this would be a worthwhile way forward, but would need more data to come up with a parametrization that is transferable between sites.
- **Length**: We have streamlined the text in several sections of the manuscript, including the introduction and discussion. We have also moved parts of the methods to the supplementary material and condensed the description of the cliff evolution in the results.
- **Figures**: We found a mistake in the color scale of figures 8, 10, 12, 14, S4, S5, S7, S9, which was supposed to go from 0 to 0.1 m/day instead of 0 to 1 m/day. We have now made the correction.

We think the manuscript has been strengthened by these revisions, but none of our main results or conclusions have changed.

We have provided detailed answers to each of these comments in this document. The comments are in black and our answers in blue and italics. The line numbers in our answer correspond to the ones in the 'track changes' version of the revision.

Thank you for your consideration of our revised manuscript, which we hope now is acceptable for publication. Please address correspondence to me at marin.kneib@wsl.ch.

Kind regards,

Marin Kneib and Co-authors

**Response to reviewer 1**

**The reviewer's comments are in black and our answers in blue and italics.**

In this study, the authors monitored the evolution of parts of the debris covered tongue of two glaciers (Langtang Glacier and 24K Glacier) during the monsoon season of the year 2019. They developed a very innovative setup to track the 3D surface changes of the surface at a weekly resolution. They use these weekly digital elevation models (DEMs), and some additional knowledge on the ice dynamics, to calculate the melt occurring from ice cliffs at very high spatial and temporal resolution. They compare their observed cliff melt rates to model simulations to decipher the controls on ice cliff melt and evolution.

I commend the authors for the impressive amount of work behind this manuscript. I think it is a very good addition to the literature, but I have some semi-major comments, and a number of specific ones. Addressing them might help clarifying some aspects of the manuscript.

We would like to thank Reviewer 1 for their very relevant and constructive comments. We agree that some clarification was necessary for the points raised and we have addressed these concerns as best as we could.

Specifically, we have:

1) Reformulated paragraph 3.5 to make the uncertainty calculation clearer. We have also added a table summarising the final uncertainty values.

2) Added explanations to the text as to why we did not use the dynamic ice cliff model and prepared additional model runs to show the comparison with the current implementation.

3) Streamlined the manuscript by shifting some of the methods and results sections to the supplementary material.

Major comments:

The uncertainty assessment (section 3.5) is generally done in a careful way. The authors did their best to evaluate honestly the uncertainty associated with their newly developed technique. However, I have some concerns with equation 4, which basically assumes that the errors are uncorrelated (independent) for each pixel. In this study, the errors should be largely correlated because they can originate from e.g. a non-perfect adjustment of the camera, or from the ice flow correction. I therefore suggest to revise this equation, and the text afterwards (L344-346). It is also not so clear, whether these uncertainties are re-used after in the text and figures, because for the figures, they refer to the standard deviation, and not to the uncertainty analysis.

**Our uncertainty analysis of the DEM differencing has shown that it could be decomposed into 1) a systematic (or correlated) error, given by the bias of elevation**

change over stable terrain and 2) a random (or uncorrelated) error, given by the standard deviation of elevation change over stable terrain (Fig. 5). We are able to show that the bias is negligible relative to the random error.

As pointed out by the Reviewer, Equation 4 only applies to the random error, and we agree that the current formulation of paragraph 3.5 was confusing in this sense. We have therefore reformulated it entirely to make these different points clearer and removed equation 4, as it was not particularly useful for everything that follows.

For simplicity, in the figures we indeed showed only the standard deviation of melt across the transects, which represents 1) the random error from the DEMs and 2) the melt variability at the surface of the cliff, and did not include the systematic uncertainty, which is much smaller, to avoid overloading the figures. We have now made this clear in the text and have added a table showing the different components of the uncertainties.

Section 3.5 was rewritten as:

[revised manuscript text omitted]

o Melt
(m) | Final
uncertainty
o Melt
(m/day) |
|----------|------------------------------------------------|--------------------------------------------------------------|----------------------------|-------------------------------------------------------------|----------------------------------|--------------------------------------------------|------------------------------------------------------|
| Langtang | 1                                              | 0.4                                                          | 2                          | 0.007                                                       | 21                               | 0.5                                              | 0.02                                                 |
| 24K      | 0.6                                            | 0.1                                                          | 1.7                        | 0.004                                                       | 14                               | 0.4                                              | 0.03                                                 |

**Table 2:** Uncertainty estimations for Langtang and 24K.

My second major comment is about the model used in the study. I don't fully understand why the authors used the static version of the model instead of the dynamic one. The dynamic model would be a great way to assess the share of each process (surface energy balance vs. 'geomorphic' processes). As it stands the manuscript is a bit frustrating, because the description of processes related to the redistribution of debris remains extremely descriptive. I am also curious about the model calibration, if there is any, because no details are provided about it.

This is a very good question, which we are happy to clarify. Indeed, the dynamicgeometry cliff model presented in Buri et al., (2016b) was extended from the staticgeometry cliff energy-balance model presented in Buri et al., (2016a) to one able to predict cliff evolution (and thus simulating changes in slope, aspect and size) by taking into account both melt and debris redistribution and neighbouring supraglacial lakes. The reviewer is correct that, ideally, a dynamic model would be appropriate for the purpose of our manuscript. However, the dynamic model was formulated and constrained with limited data of cliff geometry (a pre- and a post- ablation season DEM for each of four cliffs on one glacier). Due to the limited data available for the model development and the underlying risk of equifinality, the processes influencing the cliff dynamics (debris redistribution, additional melt from ponds) were represented by rather simple parameterisations lumping together distinct physical processes. For example, debris redistribution was simply constrained by a slope threshold above which debris would be removed, without accounting for the actual debris motion or mass conservation (Buri et al., 2016b; Moore, 2018). Similarly, the pond influence was represented by an additional melt rate at the base of the cliff, constant over the entire cliff-pond interface, and calculated from the modeling of one pond only. As such, we feel that those parameterisations and corresponding empirical parameters are too simple to represent the complexity of changes occurring during one melt season, and at the temporal scale of this study (weekly geometry updates) - whereas they were appropriate to provide bulk changes over long periods (the entire season, or monthly intervals, as in Buri et al., 2016b, 2018, 2021).

Our objective for the modelling in this study was to understand the spatiotemporal variability of the energy-balance and melt patterns of the cliffs at very high resolution. For this purpose, leveraging the actual cliff geometry at each time-step is extremely beneficial, as the actual and modelled cliff geometry will diverge even over short timescales (1 week) due to the cliff process complexity, the lumped process representation in the dynamic model discussed above, and the very high temporal resolution at which the geometry needs to be updated. As such, the dynamic model would be less reliable for understanding the local energy balance, even if calibrated to the studied cliffs. Overall, we found that the mixed approach (observations of cliff geometry to drive the energybalance model, which is the model component for which we have the highest confidence based on past validation efforts [Sakai et al., 1998, 2002; Han et al., 2010; Reid and Brock, 2014; Steiner et al., 2015]) allows to leverage the best of the observations (high resolution, frequent DEMs) and the best of the advanced model of Buri et al., 2016a (its EB component) to understand with unprecedented resolution and detail cliff melt patterns. The further development of a dynamically evolving cliff geometry model appropriate for high temporal resolution is of high interest to our group of authors and this line of research, but it would require a substantial investment to collect additional data from more cliffs, and more than two glaciers, to make the model physically representative, and it was thus outside the scope of this work - where our main goal was to understand the complex patterns of short term cliff evolution over the studied cliffs. The key step forward in our study is the ability to constrain cliff geometry changes on a weekly basis and calculate an adjusted energy balance (notably from radiative fluxes) based upon a known cliff geometry. Accordingly, we retain a high confidence that we are modelling the energy balance well at the surface.

We do feel that the development of a more process-oriented dynamic cliff model would be an important advance for the community. Advances in the past few years may contribute towards this, for instance improvements in the representation of debris motion at the surface of glaciers (Moore, 2018; 2021; Anderson and Anderson, 2018; van Woerkom et al., 2019; Westoby et al., 2020). While there are still some important knowledge gaps (for example related to the sliding of debris on steeper slopes or the debris evacuation by streams or ponds), we are convinced that the way forward for the cliff dynamic model would be to represent these processes in a much more physical way. This was not possible a few years ago, but the multitemporal UAV or time-lapse datasets (DEMs, orthoimages) that have been produced in recent years (Fyffe et al., 2020; Westoby et al., 2020; Sato et al., 2021; this study) should enable this next major step in future work.

Despite all these different elements, we understand the interest of the reviewer regarding the ability of the dynamic model to represent the melt and evolution of ice cliffs at different sites. We therefore conducted additional tests with the cliff dynamic model to show how it behaved in this particular framework (Fig. R1, R2). We ran the cliff dynamic model at 1m resolution for all four studied cliffs (tests at 1.2m resolution showed that the spatial resolution had a limited influence on the model results), updating the geometry every time there was a new time-lapse DEM (~weekly intervals). We used a slope threshold of 35° below which the cliff could be reburied and used otherwise the same parameters as the ones used in the study by Buri et al. (2016b). In all cases the cliffs shrink rapidly and disappear entirely within one month (Fig. R1, R2). We additionally ran the dynamic model with a monthly temporal resolution for the 24K cliffs, which led to the cliff shrinking, but much slower than with weekly geometry updates, as it survived three months longer (Fig. R3).

This shows that the slope threshold is not the only parameter responsible for this unexpected evolution of the cliffs, and that this is at least partly due to the high temporal frequency of the geometry updates, for which the spatial resolution is much higher than the melt distance. The dynamic model was not designed for such cases, and therefore struggled to maintain the cliffs. It would therefore require to be entirely re-calibrated to be relevant here, which is beyond the scope of this study as it does not contribute particularly to our research questions. Furthermore, based on the availability of this data, we believe that further efforts in this direction should rather aim to represent the geometry updates and debris processes in a more physical way (including updating meshed point-cloud-based surfaces rather than Eulerian grids) to make this model more transferable, rather than re-calibrating it for each specific case.

Figure R1: Initial (black) and updated cliff outlines for the Langtang cliffs at weekly time steps until full reburial of the cliffs when modelling changes in cliff dynamics following Buri et al., 2016b (a-c) and comparison with actual outlines (d-f).

Figure R2: Initial (black) and updated cliff outlines for the 24K cliff at weekly time steps until full reburial of the cliff when modelling changes in cliff dynamics following Buri et al., 2016b (a) and comparison with actual outlines (b).

Figure R3: Initial (black) and updated cliff outlines for the 24K cliff at monthly time steps until full reburial of the cliff when modelling changes in cliff dynamics following Buri et al., 2016b.

We have also indicated the different points mentioned above directly in the paper by adding the following sentences in Sections 3.7. and 5.3:

*L407-414:* 'We used the static version of the model to focus on the contribution of the different energy-fluxes only, thus removing the influence of the modeled geometry updates. Indeed, the cliff dynamic model was designed to represent changes over long periods (entire melt season or monthly intervals), for which the melt rates are high relative to the model's spatial resolution. Due to the limited data available for the model development, the processes influencing the cliff dynamics (debris redistribution, additional melt from ponds) were also represented by rather simple parametrizations lumping together distinct physical processes. While this dynamic model is appropriate to estimate bulk changes over long periods (Buri et al., 2021; Buri and Pellicciotti, 2018; Buri et al., 2016b), we considered it to be too simple to represent all the complexity of changes occurring on a weekly time-scale, and therefore less reliable to understand the local energy-balance.'

*L666-669:* 'With the growing availability of high-quality multi-temporal observations of debriscovered glacier surfaces (Westoby et al., 2020; Sato et al., 2021), including from time-lapse photogrammetry, future model developments in this direction should attempt to reconcile mechanisms of cliff backwasting that are driven primarily by the cliff energy balance with debris redistribution processes and the influence of supraglacial hydrology. '

For the descriptiveness of the text regarding debris motion, we have streamlined the results description in section 4.2 and focused on the main findings (see detailed response to comment below).

Finally, to answer the question related to the calibration of the energy-balance model, we used the exact same parameters as Buri et al., (2016a), no further calibration was conducted and the same parameters were used for both glaciers. We have specified this point in the first paragraph of section 3.7:

*L403-404:* 'We used the exact same parameters as Buri et al. (2016a) at both sites, and did not conduct any further calibration.'

My third comment is about some sentences in the discussion and conclusion that I find slightly misleading, or not well supported by the data. For instance, in I253-254, I400-403 and I631-633, it is written that time and space integrated methods lead to an underestimation of melt rates of 50%. This is not really correct, it is just that the methods are looking are measuring different targets: previous methods measured integrated losses that include reburial and expansion, and thus also melt beneath debris, while here the authors focus on shorter time scale and on melt rates along cliff transects. The same comment applies for the comparison between the melt estimates calculated for the ice cliffs vs. the sub-debris mel

---

## Author Response (AR2)

**Reply to comments on the manuscript**

**'Sub-seasonal variability of supraglacial ice cliff melt rates and associated processes from time-lapse photogrammetry'**

Marin Kneib, Evan S. Miles, Pascal Buri, Stefan Fugger, Michael McCarthy, Thomas E. Shaw, Zhao Chuanxi, Martin Truffer, Matthew J. Westoby, Wei Yang, and Francesca Pellicciotti

**Response to editor**

Dear authors,

Thank you for your detailed revisions to the manuscript. The reviewers suggest that your paper is much improved after reivision. I'm happy to accept your paper after some technical corrections.

Best,

Dr. Kang Yang
Editor, The Cryosphere

Line 38 - > should read 'Similar to supraglacial ponds, the surfaces of ice cliffs are...'

*Dear Prof. Yang,*

*Thank you very much for accepting our manuscript and for the very smooth revision process that has helped nicely improve the original manuscript. We have now corrected line 38 as suggested and uploaded all the requested documents.*

*Kind regards,*

*Marin Kneib and Co-authors*